# Surgical Fine-Tuning Improves Adaptation to Distribution Shifts

**Yoonho Lee**[*]
Stanford University

**Annie S. Chen**[*]
Stanford University

**Fahim Tajwar**
Stanford University

**Ananya Kumar**
Stanford University

**Huaxiu Yao**
Stanford University

**Percy Liang**
Stanford University

**Chelsea Finn**
Stanford University

## Abstract

A common approach to transfer learning under distribution shift is to fine-tune the last few layers of a pre-trained model, preserving learned features while also adapting to the new task. This paper shows that in such settings, selectively fine-tuning a subset of layers (which we term *surgical fine-tuning*) matches or outperforms commonly used fine-tuning approaches. Moreover, the type of distribution shift influences which subset is more effective to tune: for example, for image corruptions, fine-tuning only the first few layers works best. We validate our findings systematically across seven real-world data tasks spanning three types of distribution shifts. Theoretically, we prove that for two-layer neural networks in an idealized setting, first-layer tuning can outperform fine-tuning all layers. Intuitively, fine-tuning more parameters on a small target dataset can cause information learned during pre-training to be forgotten, and the relevant information depends on the type of shift.

## 1 Introduction

While deep neural networks have achieved impressive results in many domains, they are often brittle to even small distribution shifts between the source and target domains (Recht et al., 2019; Hendrycks & Dietterich, 2019; Koh et al., 2021). While many approaches to robustness attempt to directly generalize to the target distribution after training on source data (Peters et al., 2016; Arjovsky et al., 2019), an alternative approach is to fine-tune on a small amount of labeled target datapoints. Collecting such small labeled datasets can improve downstream performance in a cost-effective manner while substantially outperforming domain generalization and unsupervised adaptation methods (Rosenfeld et al., 2022; Kirichenko et al., 2022). We therefore focus on settings where we first train a model on a relatively large source dataset and then fine-tune the pre-trained model on a small target dataset, as a means of adapting to distribution shifts.

The motivation behind existing fine-tuning methods is to fit the new data while also preserving the information obtained during the pre-training phase. Such information preservation is critical for successful transfer learning, especially in scenarios where the source and target distributions share a lot of information despite the distribution shift. To reduce overfitting during fine-tuning, existing works have proposed using a smaller learning rate compared to initial pretraining (Kornblith et al., 2019; Li et al., 2020), freezing the early backbone layers and gradually unfreezing (Howard & Ruder, 2018; Mukherjee & Awadallah, 2019; Romero et al., 2020), or using a different learning rate for each layer (Ro & Choi, 2021; Shen et al., 2021).

We present a result in which preserving information in a non-standard way results in better performance. Contrary to conventional wisdom that one should fine-tune the last few layers to re-use the learned features, we observe that fine-tuning only the *early layers* of the network results in better performance on image corruption datasets such as CIFAR-10-C (Hendrycks & Dietterich, 2019). More specifically, as an initial finding, when transferring a model pretrained on CIFAR-10 to CIFAR-10-C by fine-tuning on a small amount of labeled corrupted images, fine-tuning only the first block of layers and freezing the others outperforms full fine-tuning on all parameters by almost 3% on average on unseen corrupted images.

---

[*]Equal contribution. Correspondence to yoonho@stanford.edu and asc8@stanford.edu.

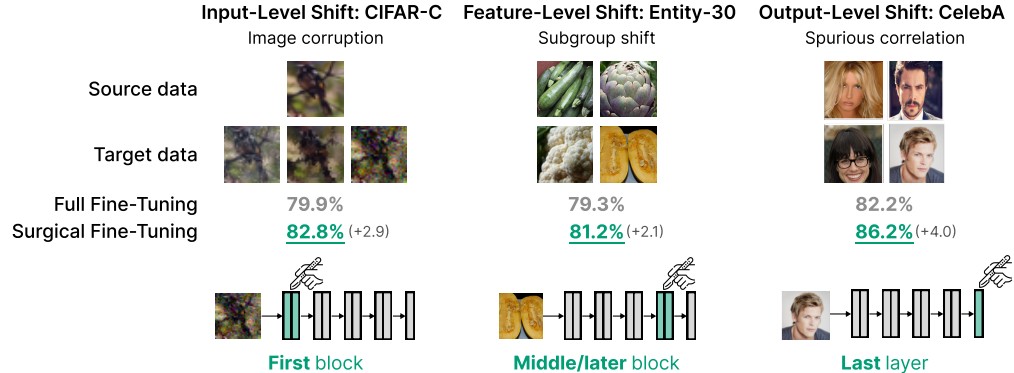

Figure 1: Surgical fine-tuning, where we tune only one block of parameters and freeze the remaining parameters, outperforms full fine-tuning on a range of distribution shifts. Moreover, we find that tuning different blocks performs best for different types of distribution shifts. Fine-tuning the first block works best for input-level shifts such as CIFAR-C (image corruption), later blocks work best for feature-level shifts such as Entity-30 (shift in entity subgroup), and tuning the last layer works best for output-level shifts such as CelebA (spurious correlation between gender and hair color).

To better understand this counterintuitive result, we study a general class of fine-tuning algorithms which we call *surgical fine-tuning*, defined as fine-tuning only a small contiguous subset of all layers in the pre-trained neural network. Equivalently, we could define surgical fine-tuning as freezing all but a few layers during fine-tuning. Parameter freezing can be beneficial because, depending on the relationship between the source and target tasks, some layer parameters trained on the source task may be close to a minima for the target distribution. Therefore, freezing these layers can facilitate generalization to the target distribution. We evaluate the performance of surgical fine-tuning with various layer choices on 7 different distribution shift scenarios, which we categorize into input-level, feature-level, and output-level shifts. As shown in Figure 1, fine-tuning only the first block of layers, the middle block, or the last layer can perform best in different distribution shift conditions, with the best such subset consistently outperforming fine-tuning all parameters.

To support our empirical results, we theoretically analyze why different types of distribution shifts require fine-tuning different layers. For two-layer neural networks, we show why fine-tuning the first layer is better for input perturbations but fine-tuning the last layer is better for label perturbations. We then present a setting where surgical fine-tuning on the first layer provably outperforms fine-tuning all parameters. If the target distribution contains only a few new "directions" (inputs outside the span of the source distribution), we show that tuning only the first layer can learn these new directions with very few target examples, while preserving all the information learned from the source distribution. However, we show that full fine-tuning forgets information learned from the source distribution—the last layer changes to accommodate the new target directions, but now performs poorly on examples outside the span of the training data. Motivated by the theoretical insight that freezing some layers can help generalization, we empirically analyze two criteria for automatically selecting layers to tune based on loss gradients. Tuning the layers selected by such criteria can also outperform full fine-tuning, though this procedure does not outperform manually choosing the best layers to tune.

Our main contribution is the empirical observation that fine-tuning only a small contiguous subset of layers can outperform full fine-tuning on a range of distribution shifts. Intriguingly, the best layers to tune differ for different distribution shift types (Figure 1). This finding is validated empirically across seven real-world datasets and three types of distribution shifts, and theoretically in an idealized two-layer neural network setup. We additionally empirically analyze two criteria for automatically selecting which layers to tune and find that fine-tuning only the layers with higher relative gradient norm outperforms full fine-tuning.

## 2 SURGICAL FINE-TUNING: FREEZING PARAMETERS DURING ADAPTATION

Our problem setting assumes two datasets from different distributions: a large dataset following the source distribution $P_{\text{src}}$, and a relatively smaller dataset following the target distribution $P_{\text{tgt}}$. The objective is to achieve high accuracy on *target data* by leveraging the different but closely related source distribution, a common scenario in real-world applications that require adaptation. For example, the source dataset can be the $50,000$ training images in CIFAR-10 (Krizhevsky et al., 2009) while the target dataset is a smaller set of 1000 corrupted CIFAR datapoints with the same image corruption (Hendrycks & Dietterich, 2019); see Figure 1 for more examples of source-target

dataset pairs that we consider. To achieve high performance on the target distribution, a model should broadly fit the large source dataset and make minor adjustments based on the smaller target dataset.

We empirically evaluate transfer learning performance with a two-stage training procedure consisting of pre-training and fine-tuning. First, we pre-train a network to minimize the loss on the source dataset to obtain $f_{\text{src}}$, which has high accuracy in the source distribution. The fine-tuning stage starts from pre-trained model parameters and minimizes the loss on the labeled target data, resulting in the model $f^{\text{tgt}}$. We evaluate two fine-tuning settings in this section: supervised fine-tuning (Section 2.1) and unsupervised adaptation (Section 2.2). In all experiments, we perform early stopping on held-out target data according to the fine-tuning loss. Finally, we evaluate the performance of the fine-tuned model on held-out data from the target distribution, i.e. $\mathcal{L}_{\text{tgt}}(f^{\text{tgt}}) = \mathbb{E}_{(x,y) \sim P_{\text{tgt}}}[\ell(f^{\text{tgt}}(x), y)]$.

Our main focus is analyzing **surgical fine-tuning**, in which we fine-tune only a subset of layers of the pre-trained model while keeping the others frozen. Denote the pre-trained model as $f = f_n \circ \ldots \circ f_1(x)$, where each layer $f_i$ has parameters $\theta_i$, and the empirical target loss as $\widehat{\mathcal{L}}_{\text{tgt}}$. Formally, surgical fine-tuning with respect to a subset $S \subseteq \{1, \ldots, n\}$ of layers is defined as solving the optimization problem

$$\arg\min_{\theta_i \ \forall i \in S} \widehat{\mathcal{L}}_{\text{tgt}}(f(\theta_1, \ldots, \theta_n)), \tag{1}$$

where all non-surgery parameters ($\theta_i$ for $i \notin S$) are fixed to their pre-trained values. Typical choices of parameters to optimize are fine-tuning all ($S = \{1, \ldots, n\}$), last ($S = \{n\}$), or the last few layers ($S = \{n - k, \cdots, n\}$). The main novelty of the surgical fine-tuning framework is that it additionally considers tuning earlier layers while keeping later layers frozen. For example, surgical fine-tuning on the first layer ($S = \{1\}$) updates only $\theta_1$, resulting in the fine-tuned model $f^{\text{tgt}}(x) = f_n^{\text{src}} \circ \ldots \circ f_2^{\text{src}} \circ f_1^{\text{tgt}}(x)$.

Intuitively, surgical fine-tuning can outperform full fine-tuning when some layers in $f^{\text{src}}$ are already near-optimal for the target distribution. As a hypothetical example, consider a scenario where there exist first-layer parameters $\theta_1^*$ such that changing only the first layer of $f_{\text{src}}$ to $\theta_1^*$ achieves zero target loss. Here, first-layer fine-tuning ($S = \{1\}$) can find $\theta_1^*$ with a small amount of target data, while full fine-tuning ($S = \{1, \ldots, n\}$) may needlessly update the other layers and thus underperform on held-out target data due to overfitting. We note that the efficacy of parameter freezing is a consequence of having limited target data, and choosing a bigger $S$ will be beneficial in settings where target data is plentiful. Now that we have introduced the problem set-up, we will next empirically investigate how surgical fine-tuning with different choices of $S$ performs on real datasets.

## 2.1 Surgical Fine-Tuning: Experiments on Real Data

In this subsection, we aim to empirically answer the following question: how does surgical parameter fine-tuning compare to full fine-tuning in terms of sample efficiency and performance on real-world datasets?

**Datasets.** We run experiments on nine real-world distribution shifts, categorized into input-level, feature-level, output-level, and natural shifts, with examples shown in Figure 1. For more details about these datasets, see Appendix B.3.

- **Input-level shift**: (1) **CIFAR-C** (Hendrycks & Dietterich, 2019), (2) **ImageNet-C** (Kar et al., 2022). The source distributions correspond to the original CIFAR-10 and ImageNet datasets (Krizhevsky et al., 2009; Deng et al., 2009), respectively. The task is to classify images from the target datasets, which consist of corrupted images.

- **Feature-level shift**: (3) **Living-17** and (4) **Entity-30** (Santurkar et al., 2020): While the source and target distributions consist of the same classes, they contain different subpopulations of those classes. For example, in Entity-30, for the class "vegetables", $P_{\text{src}}$ and $P_{\text{tgt}}$ will contain different subclasses of vegetables.

- **Output-level shift**: (5) **CIFAR-Flip**, (6) **Waterbirds**, and (7) **CelebA** (Sagawa et al., 2019). CIFAR-Flip is a synthetic task where the $P_{\text{src}}$ consists of the original CIFAR-10 dataset and the target distribution is the same dataset where each label $y$ has been flipped to be $9 - y$, e.g. the label 0 is now label 9 and vice versa. For Waterbirds and CelebA, the task labels are spuriously correlated with an attribute. The source distribution $P_{\text{src}}$ is the training set while the target distribution $P_{\text{tgt}}$ is a balanced subset with equal amounts of each of the four (label, spurious attribute) groups.

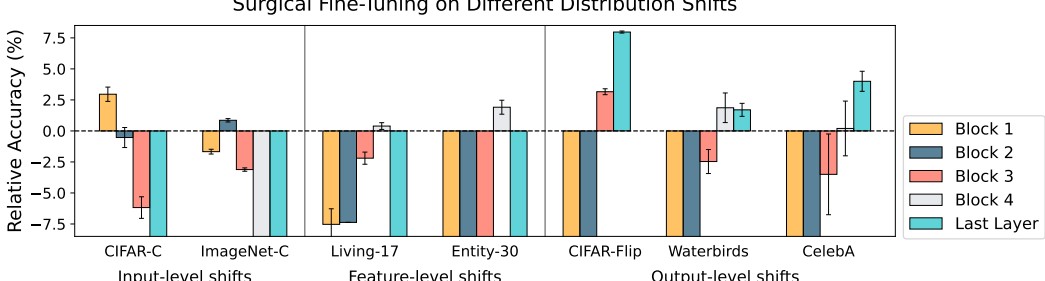

Figure 2: We present plots of relative accuracy, i.e. (surgical fine-tuning accuracy) - (full fine-tuning accuracy), along with standard errors across three runs. Fine-tuning a single parameter block can outperform full fine-tuning, and more importantly, different blocks are best for different distribution shifts. The location of the best block reflects the nature of the shift: tuning earlier layers performs best for input-level shifts while tuning later layers is best for output-level shifts.

- **Natural shift**: (8) **Camelyon17** (Bandi et al., 2018) and (9) **FMoW** (Christie et al., 2018), part of the WILDS (Koh et al., 2021) benchmark. Camelyon17 contains medical images from different hospitals where variations in data collection and processing produces naturally occurring distribution shifts across data from different hospitals. FMoW contains satellite imagery of 62 types of buildings or land use, with both temporal and sub-population shifts resulting from geographical diversity.

**Model architectures and pre-training.** For each task, before pre-training on $P_{\mathrm{src}}$, we initialize with a model with ResNet-50 architecture pre-trained on ImageNet, except for experiments with CIFAR-C and CIFAR-Flip, which both use a ResNet-26 trained from scratch, and experiments on Camelyon17 and FMoW, which use a pre-trained CLIP ViT-B/16. After initialization, we pre-train on the source domain $P_{\mathrm{src}}$ and then fine-tune on a small amount of data from the target domain. We fine-tune with the Adam optimizer, sweeping over 3 learning rates. We choose the best hyperpamters and early stop based on accuracy on held-out target data. We report results across 3 seeds for all experiments. See Appendix B.4 for more fine-tuning details.

**Surgical fine-tuning.** The models used consist of three (for ResNet-26) or four (for ResNet-50) convolutional blocks followed by a final fully connected layer. We denote these blocks as "Block 1", "Block 2", etc in the order that they process the input, and the fully connected layer as "Last Layer". CLIP ViT-B/16 has 1 embedding layer, 11 attention blocks and a linear classifier as the last layer. For each experimental setting, we report the relative target distribution accuracy and standard error across three runs after surgical fine-tuning on each block of the network, fine-tuning only that block while freezing all other parameters. We compare against full fine-tuning, i.e. tuning all parameters to minimize target loss.

**Experimental results.** Results in Figure 2 show that on every domain, surgically fine-tuning one block of the network outperforms tuning all parameters on the target distribution. We note that even matching full fine-tuning performance with surgical fine-tuning would indicate that ignoring some gradients is harmless; these results show that ignoring some gradients has a *positive effect*. Furthermore, we find that the best block to fine-tune is different across settings, depending on the nature of the distribution shift between source and target data. Datasets with an input-level shift are best handled by tuning the first network block, and similarly for feature-level shifts with a middle block and output-level shifts with the last layer. We see the a similar phenomenon in the natural shifts: Camelyon17 is closer to an input-level shift due to the lighting difference in different hospitals, while the shift between different regions in FMoW can be seen as close to a feature-level shift because building shape and spacing is most salient in satellite imagery. Quantitative results in in Table 1 are in agreement with this intuition, where tuning the earliest embedding layer works best for Camelyon17 and tuning later attention blocks works best for FMoW. Following Kumar et al. (2022b), we

| Parameters | Camelyon17 | FMoW |
|---|---|---|
| No fine-tuning | 86.2 | 35.5 |
| All | 92.3 (1.7) | 38.9 (0.5) |
| Embedding | **95.6 (0.4)** | 36.0 (0.1) |
| First three | 92.5 (0.5) | 39.8 (1.0) |
| Last three | 87.5 (4.1) | **44.9 (2.6)** |
| Last layer | 90.1 (1.5) | 36.9 (5.5) |

Table 1: OOD set accuracies after surgically fine-tuning different parameters in a CLIP ViT-B/16 model for two WILDS datasets. Bold numbers represent superior results for a dataset, and we also report the standard deviation from runs with 3 different seeds.

also evaluate fine-tuning performance with the AdamW optimizer; results in Table 8 show a similar tendency but with smaller performance gaps. In Figure 4, we find that on CIFAR-C, fine-tuning the first block matches and even outperforms full fine-tuning as well as tuning with other individual blocks when given varying amounts of data for tuning, although the gap between Block 1 and All decreases as the number of training points increases.

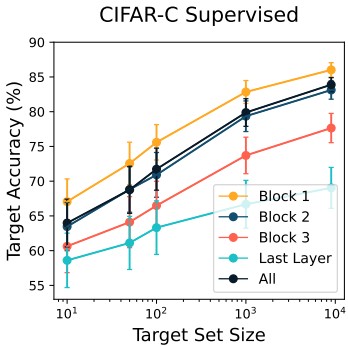

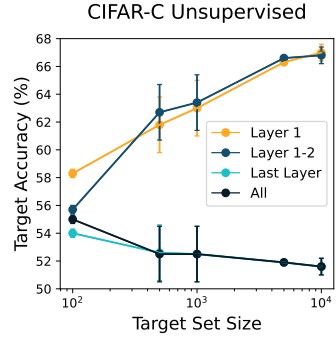

| Parameters | Episodic | Online |
|---|---|---|
| No adaptation | 67.8 | - |
| All | **71.7** | 69.7 |
| Layer 1 | 69.0 | 75.4 |
| Layer 1-2 | 69.0 | **75.5** |
| Block 1-2 | 69.0 | 75.3 |
| Last | 67.9 | 67.8 |

Table 2: Unsupervised adaptation accuracy on CIFAR-10-C, averaged across 10 corruptions.

Figure 3: Surgical fine-tuning results on CIFAR-10-C with varying amounts of target data. Fine-tuning the early layers is beneficial even in the few-shot setting, given as few as 1 image per class for tuning. Error bars indicate the average standard error over 14 corruptions.

Figure 4: Online unsupervised adaptation with surgical fine-tuning for the Gaussian corruption in the CIFAR-C dataset. Adding data in an online fashion to full or last-layer tuning results in worse performance, whereas more data helps for first-layer adaptation. Error bars indicate the standard error across three runs.

| Parameters | Episodic | Online |
|---|---|---|
| No adaptation | 46.5 | - |
| All | **47.8** | 1.8 |
| Layer 1 | 46.6 | 46.4 |
| Block 1 | 46.7 | **49.0** |
| Last | 46.5 | 46.5 |

Table 3: Unsupervised adaptation accuracy on ImageNet-C, averaged across 14 corruptions.

Intuitively, why might surgical fine-tuning match or even outperform full fine-tuning on distribution shifts? For each type of shift we consider (input-level, feature-level, and output-level), there is a sense in which one aspect of the distribution changes while everything else is kept the same, therefore requiring modification of only a small part of information learned during pre-training. For example, in image corruptions (categorized as an input-level shift), pixel-wise local features are shifted while the underlying structure of the data is the same in the source and target distributions. On the other hand, in a label shift (categorized as an output-level shift), the pixel-wise features remain the same in the source and target distributions while the mapping from final features to labels is shifted. This intuition is also in line with the independent causal mechanisms (ICM) principle (Schölkopf et al., 2012; Peters et al., 2017), which states that the causal generative process of a system's variables is composed of autonomous modules that do not inform or influence one another. From this viewpoint, distribution shifts should correspond to local changes in the causal generative process. Because discriminative models learn to invert the generative process from label to datapoint, it suffices to fine-tune only the region of the network that corresponds to the change in the causal process. We formalize this intuition more concretely in our theoretical analysis in Section 3.

## 2.2 UNSUPERVISED ADAPTATION WITH PARAMETER FREEZING

In this subsection, we aim to validate whether the findings from Section 2.1 hold in the unsupervised test-time adaptation setting, where we adapt a model trained on source data to the target distribution using only unlabeled target data. We experiment with variants of a representative state-of-the-art unsupervised adaptation method (Zhang et al., 2021a, MEMO), which minimizes marginal entropy of averaged predictions for a single image. We consider two settings: *online*, where the model retains updates from past test images, and *episodic*, where we reset the model back to the pre-trained weights after every test image.

Results in Table 2 and Table 3 show that the highest accuracy is achieved by adapting the first two layers and first block in the online setting for CIFAR-10-C and ImageNet-C respectively, and doing so outperforms fine-tuning all parameters. With full fine-tuning, online MEMO performance deteriorates as the test set size increases due to distortion of pre-trained features, as shown graphically in Figure 4. In contrast, surgical fine-tuning mitigates this effect. These results are consistent with the supervised learning experiments in Section 2.1, where adapting the early parameters was best for image corruption datasets. We show detailed results in Appendix B.6.

## 3 ANALYSIS OF SURGICAL FINE-TUNING

We now present a theoretical and empirical analysis on idealized examples of distribution shifts, to better understand the role of surgical parameter tuning in our previous experimental results. In Section 3.1, we present a setting with two-layer neural networks where tuning only the first layer can obtain zero loss on the target task while tuning only the last layer cannot and vice versa. Then, in Section 3.2, we study a setting in which tuning only the first layer provably achieves zero loss while full fine-tuning overfits and gets non-zero loss due to limited data. Finally, in Section 3.3, to support this theoretical analysis, we construct distribution shifts where localized subsets of parameters are substantially better suited for adaptation than tuning all or other parameters.

**Theoretical setup.** We focus on regression, where our goal is to map inputs $x \in \mathbb{R}^d$ to outputs $y \in R$, and $l(y, \hat{y}) = (y - \hat{y})^2$ is the squared loss. We consider two-layer networks $f_{v,B}(x) = v^\top \phi(Bx)$ where $v \in \mathbb{R}^k$, $B \in \mathbb{R}^{k \times d}$, and $\phi$ is an elementwise activation function such as ReLU. Let $x_{\text{src}}, y_{\text{src}} \sim P_{\text{src}}$ and $x_{\text{trg}}, y_{\text{trg}} \sim P_{\text{trg}}$ be the inputs and outputs in the source and target distributions. We assume $y_{\text{src}} = f_{v_{\text{src}}, B_{\text{src}}}(x_{\text{src}})$ for some $v_{\text{src}}, B_{\text{src}}$. Note that $x_{\text{src}}, y_{\text{src}}, x_{\text{trg}}, y_{\text{trg}}$ are all random variables, and expectations are taken over all random variables if not specified. We define the population losses for source and target as $L_{\text{src}}(v, B) = \mathbb{E}\big[l(f_{v,B}(x_{\text{src}}), y_{\text{src}})\big]$ and $L_{\text{trg}}(v, B) = \mathbb{E}\big[l(f_{v,B}(x_{\text{trg}}), y_{\text{trg}})\big]$.

## 3.1 Layer Choice and Expressivity: Why Fine-Tuning the Right Layer Matters

First note that for two-layer neural networks, we have two choices for surgical fine-tuning: the first layer and the last layer. We show by construction that if the distribution shift is closer to the input then first-layer tuning is better, but if the shift is closer to the output then last-layer tuning is better. In this section, we assume that $\phi$ is the elementwise ReLU function: $\phi(x)_i = \max(x_i, 0)$. Recall that we first train on lots of source data—suppose this gives us pretrained parameters $\hat{v}_{\text{src}}, \hat{B}_{\text{src}}$ which achieve minimum *source* loss: $L_{\text{src}}(\hat{v}_{\text{src}}, \hat{B}_{\text{src}}) = 0$.

**Input perturbation.** Suppose that the target input is a "perturbed" or "corrupted" version of the source input: $x_{\text{trg}} = Ax_{\text{src}}$ for some invertible matrix $A \in \mathbb{R}^{n \times n}$, where the corresponding label is unchanged: $y_{\text{trg}} = y_{\text{src}}$. We note that this simplified perturbation class includes some common image corruptions as brightness shift and Gaussian blur as special cases, while others such as pixelation are similarly linear projections but non-invertible. Proposition 1 shows that for this distribution shift, tuning only the first layer can minimize the target loss but only changing the last layer may not.

**Proposition 1.** *For all $A, P_{\text{src}}, P_{\text{trg}}$ with $x_{\text{trg}} = Ax_{\text{src}}$ for invertible $A$ and $y_{\text{trg}} = y_{\text{src}}$, there exists a first-layer $B$ that can minimize the target loss: $\min_B L_{\text{trg}}(\hat{v}_{\text{src}}, B) = 0$. However, changing the last layer may not be sufficient: there exists such $A, P_{\text{src}}, P_{\text{trg}}$ such that the target loss is non-zero for any choice of last layer $v$: for all $i$, $\min_v L_{\text{trg}}(v, \hat{B}_{\text{src}}) > 0$.*

Intuitively, the first-layer can learn to "undo" the perturbation by selecting $B = A^{-1}$. However, if we freeze the first-layer then the representations $\phi(\hat{B}_{\text{src}} x_{\text{trg}})$ may miss important input directions in the target, so no last-layer $v$ can produce the correct output. For a full statement and proof, see Appendix B.1.

**Label perturbation.** Now suppose that the source and target inputs are the same $x_{\text{trg}} = x_{\text{src}}$, but the target output is perturbed from the source output: $y_{\text{trg}} = t y_{\text{src}}$ for some $t$. Proposition 2 shows that tuning only the first layer may not achieve non-zero target loss for this distribution shift while tuning the last layer will do so.

**Proposition 2.** *For all $t, P_{\text{src}}, P_{\text{trg}}$ with $x_{\text{trg}} = x_{\text{src}}$, and $y_{\text{trg}} = t y_{\text{src}}$, there exists a last-layer $v$ that can minimize the target loss: $\min_v L_{\text{trg}}(v, \hat{B}_{\text{src}}) = 0$. However, changing the first layer may not be sufficient—there exists such $t, P_{\text{src}}, P_{\text{trg}}$ such that the target loss is non-zero for any choice of first layer $B$: $\min_B L_{\text{trg}}(\hat{v}_{\text{src}}, B) = 0$*

Similarly to Proposition 1, the last layer can adapt to the label shift by "reversing" the multiplication by $t$. In contrast, when the last layer is frozen and only the first layer is tuned, we may lack expressivity due to the information destroyed by the ReLU activation $\phi(\cdot)$. For a full statement and proof, see Appendix B.1.

## 3.2 Can Surgical Fine-Tuning Outperform Full Fine-Tuning?

In this section, we show that first-layer fine-tuning can provably outperform full fine-tuning when we have an insufficient amount of target data. We show that this can happen even in tuning two-layer linear networks (Kumar et al., 2022a), where $\phi$ is the identity map: for all $i$, $\phi(x)_i = x_i$. Our analysis suggests perhaps a more general principle underlying the benefits of surgical fine-tuning over full fine-tuning: by fine-tuning more parameters than necessary, the model can overfit to the small target dataset while forgetting relevant information learned during pre-training.

**Pretraining.** We first start with $v_0, B_0$ which are initialized randomly: $v_{0i} \sim N(0, \sigma_v^2)$ and $B_{0ij} \sim N(0, \sigma_B^2)$ for all $i, j$. We then run gradient descent on $L_{\text{src}}(v, B)$ to obtain $\hat{v}_{\text{src}}, \hat{B}_{\text{src}}$, which we assume minimizes the source loss: $L_{\text{src}}(\hat{v}_{\text{src}}, \hat{B}_{\text{src}}) = 0$.

|  | | Added Noise | | | |
| --- | --- | --- | --- | --- | --- |
|  | | First block | Second block | Third block | Last layer |
| **Tuned Layers** | First block | 92.5 (0.3) | 27.7 (1.2) | 27.2 (1.7) | 37.4 (9.0) |
|  | Second block | 68.8 (2.3) | 92.6 (0.3) | 46.6 (2.6) | 61.0 (4.4) |
|  | Third block | 48.0 (1.3) | 59.7 (0.6) | 92.1 (0.2) | 92.0 (0.5) |
|  | Last layer | 33.7 (0.4) | 34.8 (0.4) | 62.0 (1.1) | 91.8 (0.6) |
|  | All | 87.8 (1.2) | 91.9 (0.2) | 92.0 (0.1) | 91.3 (0.3) |

Figure 5: In this setting, we construct distribution shifts such that a particular block of parameters is substantially more suited for adaptation. We find that tuning only the subset of parameters that is responsible for the shift performs better than tuning any other block of parameters or all parameters. Darker blue indicates higher accuracy while darker red indicates lower accuracy.

**Fine-tuning data.** Suppose we have $n$ target datapoints sampled from $P_{\text{trg}}$: $(x_{\text{trg}}^{(1)}, y_{\text{trg}}^{(1)}) \ldots,$ $(x_{\text{trg}}^{(n)}, y_{\text{trg}}^{(n)})$. The empirical fine-tuning loss is given by: $\hat{L}_{\text{trg}}(v, B) = \sum_{i=1}^{n} l(f_{v,B}(x_{\text{trg}}^{(i)}), y_{\text{trg}}^{(i)})$.

**Fine-tuning algorithms.** We study two different gradient flows each corresponding to fine-tuning methods: first-layer tuning (fl) and full fine-tuning (ft).

$$\partial_t B_{\text{fl}}(t) = -\nabla_B \hat{L}_{\text{trg}}(v_{\text{fl}}(t), B_{\text{fl}}(t)) \quad \partial_t v_{\text{fl}}(t) = 0$$

$$\partial_t B_{\text{ft}}(t) = -\nabla_B \hat{L}_{\text{trg}}(v_{\text{ft}}(t), B_{\text{ft}}(t)) \quad \partial_t v_{\text{ft}}(t) = -\nabla_v \hat{L}_{\text{trg}}(v_{\text{ft}}(t), B_{\text{ft}}(t)),$$

with initial conditions $v_{\text{fl}}(0) = v_{\text{ft}}(0) = \hat{v}_{\text{src}}$ and $B_{\text{fl}}(0) = B_{\text{ft}}(0) = \hat{B}_{\text{src}}$. We denote the limit points of these gradient flows as $v_{\text{fl}}^{\infty} = \lim_{t \to \infty} v_{\text{ft}}(t)$, etc.

The following theorem shows that there exists a shift, where if we have a small target dataset, full fine-tuning does worse than first-layer tuning.

**Theorem 1.** *For any $\delta > 0$, there exists $d, k, P_{\text{src}}, P_{\text{trg}}, n$ such that with probability at least $1 - \delta$, first-layer tuning gets $0$ loss at convergence, but full fine-tuning gets higher (non-zero) loss throughout the fine-tuning trajectory:*

$$L_{\text{trg}}(v_{\text{ft}}(t), B_{\text{ft}}(t)) > L_{\text{trg}}(v_{\text{fl}}^{\infty}, B_{\text{fl}}^{\infty}) = 0 \quad \forall t. \tag{2}$$

Intuitively, if $P_{\text{ood}}$ contains a few additional directions in the input that are not present in $P_{\text{id}}$, then first layer-tuning can quickly learn those new directions. Full fine-tuning changes both the head $v$ and feature extractor $B$ to fit these new directions—however, because the head $v$ has changed it may be incompatible with $B$ in some directions not seen in the finite training set, thus "forgetting" some knowledge present in the source data. The full proof is in Appendix B.2.

### 3.3 SURGICAL FINE-TUNING ON SYNTHETIC DISTRIBUTION SHIFTS

To better illustrate how specific subsets of parameters are better suited depending on the distribution shift, we model distribution shifts by adding noise to individual blocks of layers. More specifically, we initialize with a ResNet-26 model pretrained on the CIFAR-10 (Krizhevsky et al., 2009) training dataset. We then add noise to each of the three blocks or the last layer, simulating distribution shifts localized to those parameters, and then tune each of the different blocks of the network while freezing all other parameters on the CIFAR-10 test dataset.

In Figure 5, we find that only tuning the subset of parameters that is responsible for the shift performs better than tuning any other subset of parameters and even outperforms tuning all layers, indicating that when tuning the parameters responsible for the shift, *tuning other parameters may actually hurt performance*.

## 4 AUTOMATICALLY SELECTING WHICH LAYERS TO TUNE

In this section, we investigate three criteria for automatically finding an adequate subset of layers to perform surgical fine-tuning on. We evaluate their fine-tuning performance versus full fine-tuning and another prior method on the 7 real-data domains introduced in Section 2.1. We also analyze performance on the synthetic distribution shifts introduced in Section 3.3.

### 4.1 CRITERIA FOR SELECTING LAYERS

We consider three metrics for automatically choosing which layers to freeze.

**Cross-Val**. After running surgical fine-tuning for all blocks, we select the best block based on a held-out validation set from the target distribution. While quite effective, this method requires as many fine-tuning runs as there are blocks inside the network.

| Method | Input-level Shifts | | Feature-level Shifts | | Output-level Shifts | | | Avg Rank |
|---|---|---|---|---|---|---|---|---|
| | CIFAR-C | IN-C | Living-17 | Entity-30 | CIFAR-F | Waterbirds | CelebA | |
| No Adaptation | 52.6 (0) | 18.2 (0) | 80.7 (1.8) | 58.6 (1.1) | 0 (0) | 31.7 (0.3) | 27.8 (1.9) | - |
| Cross-Val | 82.8 (0.6) | 51.6 (0.1) | 93.2 (0.3) | 81.2 (0.6) | 93.8 (0.1) | 89.9 (1.2) | 86.2 (0.8) | - |
| Full Fine-Tuning (All) | 79.9 (0.7) | 50.7 (0.1) | 92.8 (0.7) | 79.3 (0.6) | 85.9 (0.4) | **88.0 (1.2)** | 82.2 (1.3) | 2.71 |
| Gradual Unfreeze (First $\rightarrow$ Last) | 80.5 (0.7) | 50.7 (0.1) | 90.1 (0.1) | 69.9 (3.0) | 81.9 (0.6) | 87.5 (0.2) | 71.9 (0.7) | 4.71 |
| Gradual Unfreeze (Last $\rightarrow$ First) | 78.8 (0.8) | 49.8 (0.2) | 90.6 (0.2) | 77.8 (0.5) | 84.3 (0.9) | 87.8 (0.1) | 78.5 (2.9) | 4.0 |
| $L_1$ Regularize (Xuhong et al., 2018) | **81.7 (0.6)** | 48.8 (0.3) | 93.4 (0.5) | 78.4 (0.1) | 84.2 (1.2) | 87.6 (1.9) | **82.6 (1.8)** | 3.28 |
| Auto-SNR | 80.9 (0.7) | 49.9 (0.2) | **93.5 (0.2)** | 77.3 (0.3) | 17.3 (0.7) | 86.3 (0.7) | 78.5 (1.8) | 4.14 |
| Auto-RGN | 81.4 (0.6) | **51.2 (0.2)** | **93.5 (0.3)** | **80.6 (1.2)** | **87.7 (2.8)** | **88.0 (0.7)** | 82.2 (2.7) | **1.29** |

Table 4: We report the average accuracy and standard error achieved on the target distribution on 7 real-data tasks. Cross-Val, which requires as a surgical fine-tuning run for each block, performs the best, but we find that Auto-RGN performs the best out of all methods that require only 1 fine-tuning run, outperforming Full Fine-tuning, Gradual Unfreezing, $L_1$ Regularize, and Auto-SNR. The best overall method for each shift is underlined, and the best among methods that use 1 fine-tuning run is bolded.

**Relative Gradient Norm (Auto-RGN).** Within each layer, we measure the ratio of gradient norm to parameter norm, and select layers that have relatively larger gradients. Intuitively, our hypothesis is that layers with large gradient magnitudes may carry more information about the target task than others and can therefore be more useful. Formally, denote gradients at layer $i$ as $g_i$. We define the relative gradient norm of this layer as $\text{RGN}(\theta_i) = \frac{(g_i)}{||\theta_i||}$. Then to alter fine-tuning with this criterion, at each epoch, we normalize the RGNs for each layer between 0 and 1 and then multiply the learning rate for each layer by its RGN. Using this criterion for fine-tuning requires no additional hyperparameters over tuning all layers and only one fine-tuning run.

**Signal-to-Noise Ratio (Auto-SNR).** For each layer $i$, this criterion is defined as $\text{SNR}(g_i) = \frac{\text{Avg}(g_i)^2}{\text{Var}(g_i)}$, with average and variance computed across (target) datapoints. Intuitively, SNR measures how noisy the gradient of each layer is and thus how much it may contribute to distorting the function learned during pre-training. This gradient-based criterion has been shown to be useful for early stopping (Mahsereci et al., 2017). During fine-tuning, we normalize the SNR for each layer between 0 and 1 and then freeze all layers that have SNR under a threshold that is tuned as an additional hyperparameter.

As points of comparison, we compare the three criteria for layer selection above to existing methods for regularizing fine-tuning. We consider two variations of gradual unfreezing: **Gradual Unfreeze (First $\rightarrow$ Last)** and **Gradual Unfreeze (Last $\rightarrow$ First)** (Howard & Ruder, 2018; Romero et al., 2020; Kumar et al., 2022a), in addition to **$L_1$ Regularize** (Xuhong et al., 2018). These methods are similar in spirit to surgical fine-tuning in that they aim to minimize changes to parameters. We additionally experimented with regularizing the $L_2$ norm, but found that $L_1$ consistently performs better.

## 4.2 RESULTS ON REAL WORLD DATASETS

In Table 4, we compare Cross-Val with 6 methods (Full Fine-Tuning, Gradual Unfreeze (First $\rightarrow$ Last), Gradual Unfreeze (Last $\rightarrow$ First), $L_1$ Regularize, Auto-SNR, and Auto-RGN) that require only 1 fine-tuning run. We find that auto-tuning with relative grad norm (Auto-RGN) matches or outperforms fine-tuning all parameters on all domains and is the most competitive method that requires only 1 fine-tuning run although it does not quite match the performance of Cross-Val. We find that Cross-Val corresponds in performance to the best surgical fine-tuning result for each dataset, which is expected, as the validation and test sets are both held-out subsets of the same target distribution. Auto-SNR struggles to extract the most effective layers for tuning and hence does worse on most shifts than All and Auto-RGN. While Gradual Unfreeze fails to consistently outperform full fine-tuning, the directionality of results is consistent with surgical fine-tuning: unfreezing first layers is best in input-level shifts and unfreezing last layers is best in output-level shifts. $L_1$ Regularize performs slightly better than fine-tuning all, but performs worse than Auto-RGN on all datasets except CIFAR-C. All methods outperform no adaptation.

## 4.3 AUTOMATIC SELECTIVE FINE-TUNING IN SYNTHETIC DISTRIBUTION SHIFTS

As Auto-RGN is the best performing method that requires only one fine-tuning run, and in particular outperforms fine-tuning all layers, we further analyze what layers Auto-RGN chooses to fine-tune and see to what extent they correlate with our experiments in Section 2. To do so, we evaluate Auto-RGN on the synthetic distribution shifts introduced in Section 3.3, where we model distribution shifts by adding noise to blocks of parameters, and plot the weights that Auto-RGN gives to the layers.

| Tuned Layers | Full Fine-Tuning | Auto-RGN |
|---|---|---|
| Block 1 | 87.8 (1.2) | **92.0 (0.4)** |
| Block 2 | 91.9 (0.2) | **92.7 (0.4)** |
| Block 3 | 92.0 (0.1) | **92.5 (0.4)** |
| Last Layer | 91.3 (0.3) | **92.8 (0.3)** |

Table 5: Automatically choosing which layers to tune using the relative gradient norms (RGN) of each layer outperforms full fine-tuning on distribution shifts constructed by adding noise to different blocks of layers.

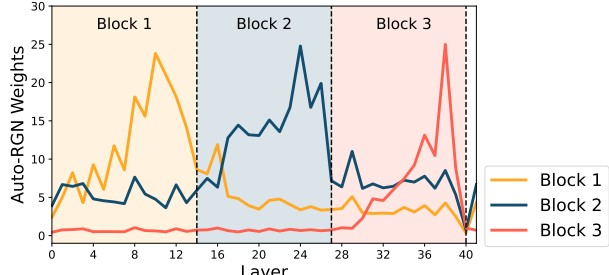

Figure 6: Auto-RGN consistently gives higher weights for the layers in the block responsible for the distribution shift than for the other layers.

We find that Auto-RGN is able to ascertain which parameters may be responsible for the shift and weight the learning of those parameters to be higher than the others, resulting in an informative signal that matches the performance of tuning only the noisy subset of parameters and outperforms full fine-tuning, as seen in Table 5. Figure 6 shows the accumulated weights given by Auto-RGN over the course of training for each layer, colored by block. The weights for the layers responsible for the distribution shifts are higher than the weights for the other layers.

## 5 RELATED WORK

**Parameter freezing.** Freezing parameters to preserve previously learned information has been shown to be an effective strategy in a diverse set of domains: domain adaptation (Sener et al., 2016; Long et al., 2016), early stopping (Mahsereci et al., 2017), generative models (Mo et al., 2020), and gradient-based meta-learning (Zintgraf et al., 2019; Raghu et al., 2019; Triantafillou et al., 2021), A highly effective approach to fast adaptation of large language models is prompt tuning (Li & Liang, 2021; Hambardzumyan et al., 2021; Lester et al., 2021; Wei et al., 2021), which can similarly be seen as an extreme special case of freezing where we only fine-tune the inputs to the neural network. Our surgical fine-tuning framework contains many such previous works as special cases, and our experiments highlight the value of carefully choosing the subset of parameters to freeze.

**Transfer learning.** Prior works in transfer learning have studied how fine-tuning may be used to adapt pretrained features to a target distribution (Oquab et al., 2014; Yosinski et al., 2014; Sharif Razavian et al., 2014). To preserve information obtained during pre-training, many works propose methods of regularizing the fine-tuning process (Zhang et al., 2020; Xuhong et al., 2018; Lee et al., 2019a; Jiang et al., 2019; Li et al., 2020; Aghajanyan et al., 2020; Gouk et al., 2021; Shen et al., 2021; Karani et al., 2021). In particular, many works show that freezing some parameters in the pre-trained model can reduce overfitting during fine-tuning (Kirkpatrick et al., 2017; Lee et al., 2019b; Guo et al., 2019; Ramasesh et al., 2020; Liu et al., 2021b; Royer & Lampert, 2020; Eastwood et al., 2021; Evci et al., 2022; Eastwood et al., 2022; Cohen et al., 2022; Touvron et al., 2022), and we build on such observations. Module criticality (Zhang et al., 2019; Chatterji et al., 2019; Neyshabur et al., 2020), which independently examines each layers' loss surface, is also closely related to our analysis. In contrast to existing works, we make the counterintuitive observation that freezing the later layers, or equivalently performing surgical fine-tuning on the early layers, can perform best in some settings. Furthermore, we study the relationship between the best subset of layers to tune and the nature of the distribution shift between the source and target distributions.

**Distribution shifts.** Many existing works have studied adaptation and robustness to various distribution shifts (Tzeng et al., 2014; Byrd & Lipton, 2019; Hendrycks et al., 2019; Arjovsky et al., 2019; Salman et al., 2020; Liu et al., 2021a; Wiles et al., 2021; Andreassen et al., 2021; Miller et al., 2021; Creager et al., 2021; Lee et al., 2022; Kumar et al., 2022a). Such works typically frame robustness to distribution shift as a zero-shot generalization problem, where the model is trained on source and evaluated on target. We consider a different problem setting where the model is allowed to adapt to some labeled target data available. Some recent works have proposed methods for model adaptation at test time (Sun et al., 2020; Varsavsky et al., 2020; Iwasawa & Matsuo, 2021; Wang et al., 2020; Zhang et al., 2021a;b; Gandelsman et al., 2022). Recent works (Rosenfeld et al., 2022; Kirichenko et al., 2022) study a problem setting close to ours, showing that fine-tuning the last layer is sufficient for adapting to datasets with a spuriously correlated attribute. Our experiments in Section 2 confirm these results, and we further evaluate on a broader set of distribution shifts including image corruptions and shifts at the level of intermediate features. We find that fine-tuning different subsets of layers performs best for different types of distribution shifts, and also present theoretical analysis on the relationship between surgical fine-tuning and the type of distribution shift.

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

## A   EXTENDED DISCUSSION

In this paper, we empirically find that when fine-tuning on a new target distribution, it is often best to perform surgical fine-tuning, i.e. to adapt only a small contiguous subset of parameters. More importantly, which subset is more effective to tune depends on the type of distribution shift. For example, on input-level shifts like image corruption, only tuning earlier layers can outperform fine-tuning all or only later layers. These results support our intuition from the independent causal mechanisms (ICM) principle: many distribution shifts can be explained by a shift in one module of the prediction mechanism and can thus be adapted to by tuning only a small subset of the network. Our empirical findings are supported by theoretical results, which show by construction that first-layer tuning may outperform full fine-tuning in an idealized two-layer neural network setting.

Additionally, manually choosing which layers to freeze with the framework of surgical fine-tuning requires more fine-tuning runs than fine-tuning all layers, so we analyze two criteria for automatically selecting which layers to tune. While Auto-RGN consistently improves over full fine-tuning, its performance does not match the best surgical fine-tuning approach. Future work may close this gap by investigating more effective criteria for automatic selection. More generally, a potentially fruitful direction for future work is in better understanding when a distribution shift would prefer a certain layer, potentially shedding light on the nature of different distribution shifts.

## B   APPENDIX

### B.1   PROOFS FOR SECTION 3.1

**Proposition 1.** *For all $A, P_{\text{src}}, P_{\text{trg}}$ with $x_{\text{trg}} = Ax_{\text{src}}$ for invertible $A$ and $y_{\text{trg}} = y_{\text{src}}$, there exists a first-layer $B$ that can minimize the target loss: $\min_B L_{\text{trg}}(\hat{v}_{\text{src}}, B) = 0$. However, changing the last layer may not be sufficient: there exists such $A, P_{\text{src}}, P_{\text{trg}}$ such that the target loss is non-zero for any choice of last layer $v$: for all $i$, $\min_v L_{\text{trg}}(v, \hat{B}_{\text{src}}) > 0$.*

*Proof.* Let $\hat{B}_{\text{src}}, \hat{v}_{\text{src}}$ be minimum loss solutions so that $y_{\text{src}} = \hat{v}_{\text{src}}\phi(\hat{B}_{\text{src}}x_{\text{src}})$ for all $x_{\text{src}}, y_{\text{src}}$. Denoting $B = \hat{B}_{\text{src}}A^{-1}$, we have for all $x_{\text{trg}}$

$$\hat{v}_{\text{src}}\phi(Bx_{\text{trg}}) = \hat{v}_{\text{src}}\phi(\hat{B}_{\text{src}}A^{-1}Ax_{\text{src}}) = \hat{v}_{\text{src}}\phi(\hat{B}_{\text{src}}x_{\text{src}}) = y_{\text{src}} = y_{\text{trg}}. \tag{3}$$

Therefore, this pair of parameters $v, B$ achieves $L_{\text{trg}}(\hat{v}_{\text{src}}, B) = 0$.

We construct a counterexample showing the impossibility of last-layer tuning as follows. Recall that $\phi(\cdot)$ is the elementwise ReLU function. Let $A = -I$, an invertible diagonal matrix with all entries $-1$. Let the source distribution be so that $\hat{B}_{\text{src}}x_{\text{src}}$ has only positive entries for all $x_{\text{src}}$ in its support. Then for any $v$, we have $v\phi(\hat{B}_{\text{src}}x_{\text{trg}}) = v\phi(-\hat{B}_{\text{src}}x_{\text{src}}) = 0$, so the expected loss is positive. Therefore, $\min_v L_{\text{trg}}(v, \hat{B}_{\text{src}}) > 0$. □

**Proposition 2.** *For all $t, P_{\text{src}}, P_{\text{trg}}$ with $x_{\text{trg}} = x_{\text{src}}$, and $y_{\text{trg}} = ty_{\text{src}}$, there exists a last-layer $v$ that can minimize the target loss: $\min_v L_{\text{trg}}(v, \hat{B}_{\text{src}}) = 0$. However, changing the first layer may not be sufficient—there exists such $t, P_{\text{src}}, P_{\text{trg}}$ such that the target loss is non-zero for any choice of first layer $B$: $\min_B L_{\text{trg}}(\hat{v}_{\text{src}}, B) > 0$*

*Proof.* Let $\hat{B}_{\text{src}}, \hat{v}_{\text{src}}$ be minimum loss solutions so that $y_{\text{src}} = \hat{v}_{\text{src}}\phi(\hat{B}_{\text{src}}x_{\text{src}})$ for all $x_{\text{src}}, y_{\text{src}}$. Let $v = t\hat{v}_{\text{src}}$. Then for all $x_{\text{trg}}$,

$$v\phi(\hat{B}_{\text{src}}x_{\text{trg}}) = t\hat{v}_{\text{src}}\phi(\hat{B}_{\text{src}}x_{\text{src}}) = ty_{\text{src}} = y_{\text{trg}}. \tag{4}$$

Therefore, this pair of parameters $v, \hat{B}_{\text{src}}$ achieves $L_{\text{trg}}(v, \hat{B}_{\text{src}}) = 0$.

We next construct a counterexample showing that tuning only the first layer may not be sufficient. Recall that $\phi(\cdot)$ is the elementwise ReLU function. Let $t = -1$. Let the source distribution be so that both $Bx_{\text{src}}$ and $\hat{v}_{\text{src}}$ consist only of positive entries for all $x_{\text{src}}$ in its support. For any $B$, both $\hat{v}_{\text{src}}$ and $\phi(Bx_{\text{trg}})$ consist only of positive entries, so $\hat{v}_{\text{src}}\phi(Bx_{\text{trg}})$ cannot express $y_{\text{trg}} = -y_{\text{src}} < 0$. so the expected loss is positive for any $B$. Therefore, $\min_B L_{\text{trg}}(\hat{v}_{\text{src}}, B) > 0$. □

### B.2   PROOF OF THEOREM 1 IN SECTION 3.2

We introduce some additional setup, prove two key lemmas which bound the loss of first-layer tuning and full fine-tuning respectively, and then this immediately implies Theorem 1.

**Defining the label distribution.** We assume that $P(y \mid x)$ is the same in both the source and the target. That is, we assume there exists some $v_*, B_*$ such that $y = v_*^\top B_* x$ for both $P_{\text{src}}$ and $P_{\text{trg}}$. Let $w_* = B_*^\top v_*$.

**Defining the covariate distributions.** Now, we define the distribution over the inputs $x$. Let the source distribution $P_{\text{src}}$ have density on a $d_{\text{src}}$-dimensional subspace, where $d_{\text{src}} < d$ (recall that the input dimension is $d$). Formally, this means that there exists some $S_{\text{src}} \in \mathbb{R}^{d \times d_{\text{src}}}$ with linearly independent columns, and some distribution $P_{\text{src}}^{(z)}$ with density on $\mathbb{R}^{d_{\text{src}}}$, such that $P_{\text{src}}$ has the distribution $S_{\text{src}} z$ where $z \sim P_{\text{src}}^{(z)}$.

We assume a non-degeneracy condition—that the optimal model $w_*$ does not map (non-zero) source examples to 0: for all source examples $x \in \text{colspace}(S_{\text{src}})$, if $x \neq 0$, then $w_*^\top x \neq 0$. If $w_*$ were random or had some noise then this would hold with probability 1.

Suppose we have an orthogonal distribution $P_{\text{orth}}$ which has density on a $d_{\text{orth}}$ dimensional subspace. Formally, this means that there exists some $S_{\text{orth}} \in \mathbb{R}^{d \times d_{\text{orth}}}$ with linearly independent columns, and some distribution $P_{\text{orth}}^{(z)}$ with density on $\mathbb{R}^{d_{\text{orth}}}$, such that $P_{\text{orth}}$ has the distribution $S_{\text{orth}} z$ where $z \sim P_{\text{orth}}^{(z)}$. We assume that the support of $P_{\text{orth}}$ and $P_{\text{src}}$ are orthogonal—that is, the columns of $S_{\text{orth}}$ and the columns of $S_{\text{src}}$ are all orthogonal.

The target distribution $P_{\text{trg}}$ is an equal mixture of the source distribution $P_{\text{src}}$ and the orthogonal distribution $P_{\text{orth}}$:

$$P_{\text{trg}} = \frac{1}{2}(P_{\text{src}} + P_{\text{orth}}) \tag{5}$$

This means that to sample from $P_{\text{trg}}$, with probability 0.5 we pick a sample from $P_{\text{src}}$ and with probability 0.5 we pick a sample from $P_{\text{orth}}$.

**First layer tuning gets 0 target loss.** We first show that first-layer tuning gets 0 loss on the target distribution:

**Lemma 1.** *For any $\delta > 0$, suppose $n > 10 d_{\text{orth}} \log \frac{2}{\delta}$. Then with probability at least $1 - \delta$, first-layer tuning gets 0 loss at convergence:*

$$L_{\text{trg}}(v_{\text{fl}}^\infty, B_{\text{fl}}^\infty) = 0 \tag{6}$$

*Proof.* We first note that first-layer tuning does not update the head $v$, so we have $v_{\text{fl}}^\infty = \hat{v}_{\text{src}}$.

**Convex so converges.** Note that the loss $\hat{L}_{\text{trg}}$ is convex in $B$. To see this, note that we can write:

$$\hat{L}_{\text{trg}}(v, B) = \sum_{i=1}^{n} (v^\top B x_{\text{trg}}^{(i)} - y_{\text{trg}}^{(i)})^2 = (\text{Tr}(x_{\text{trg}}^{(i)} v^\top B) - y_{\text{trg}}^{(i)})^2. \tag{7}$$

$\text{Tr}(x_{\text{trg}}^{(i)} v^\top B)$ is a linear function of $B$, so this is simply a least squares regression problem and is convex. This means that gradient flow converges to *a* minimizer of the *train* loss:

$$\hat{L}_{\text{trg}}(\hat{v}_{\text{src}}, B_{\text{fl}}^\infty) \leq \hat{L}_{\text{trg}}(\hat{v}_{\text{src}}, B) \quad \forall B. \tag{8}$$

However, since $\hat{v}_{\text{src}} \neq 0$, there exists $B$ such that $\hat{L}_{\text{trg}}(\hat{v}_{\text{src}}, B) = 0$, so since the loss is non-negative, this implies that:

$$\hat{L}_{\text{trg}}(\hat{v}_{\text{src}}, B_{\text{fl}}^\infty) = 0. \tag{9}$$

**Define ID and orthogonal training examples.** We note that every example $x$ sampled from $P_{\text{trg}}$ comes from exactly one of $P_{\text{orth}}$ or $P_{\text{src}}$.[*] We group examples based on which distribution they come from. Let $X_{\text{src}}$ and $X_{\text{orth}}$ denote the source and orthogonal examples respectively, where $X_i$ denotes

$$X_{\text{src}} = \{x_{\text{trg}}^{(i)} : x_{\text{trg}}^{(i)} \in \text{colspace}(S_{\text{src}})\} \quad X_{\text{orth}} = \{x_{\text{trg}}^{(i)} : x_{\text{trg}}^{(i)} \in \text{colspace}(S_{\text{orth}})\} \tag{10}$$

**Enough to get a basis correct.** Since we are working with linear models, it suffices to get all examples in a basis correct to get the entire subspace correct. Stated formally, if $v^\top B x = v_*^\top B_* x$ and $v^\top B x' = v_*^\top B_* x'$, then the equality holds for any linear combination as well: $v^\top B(\alpha x + \beta x') = v_*^\top B_* (\alpha x + \beta x')$ for all $\alpha, \beta$. So to show that $L_{\text{trg}}(\hat{v}_{\text{src}}, B_{\text{fl}}^\infty) = 0$ is suffices to show that $\hat{v}_{\text{src}}^\top B_{\text{fl}}^\infty x = v_*^\top B_* x$ for some set of $x$ that spans the support of $P_{\text{src}}$ and $P_{\text{orth}}$.

---

[*] Since the distributions have density on some subspace, $x$ is almost surely non-zero.

$X_{\text{orth}}$ **spans orthogonal subspace.** A standard application of Hoeffding gives us that with probability $\geq 1 - \delta/2$, we have at least $d_{\text{orth}}$ examples from the orthogonal distribution: $|X_{\text{orth}}| \geq d_{\text{orth}}$. Since $P_{\text{orth}}$ has density on a $d_{\text{orth}}$ dimensional subspace, from e.g., Lemma 3 in Xie et al. (2021) these examples will span the orthogonal subspace almost surely: $\text{span}(X_{\text{orth}}) = \text{colspace}(S_{\text{orth}})$. Intuitively, since $P_{\text{orth}}$ has density we will sample points in different directions.

**Get all examples in orthogonal subspace correct.** Since we have $0$ training loss, and $X_{\text{orth}}$ spans the support of $P_{\text{orth}}$, this means that we get all examples in the orthogonal subspace correct—that is, for all $x$ in the support of $P_{\text{orth}}$, we have: $\hat{v}_{\text{src}}^\top B_{\text{fl}}^\infty x = v_*^\top B_* x$.

**Get all examples in source subspace correct.** For the source subspace we will split into two cases. First, we define the region of the source subspace $\text{colspace}(S_{\text{src}})$ that is orthogonal to all source training examples $X_{\text{src}}$:

$$X_{\text{src}}^\perp = \{x \in \text{colspace}(S_{\text{src}}) : \forall x' \in X_{\text{src}}.\ x \perp x'\}. \tag{11}$$

Since we have $0$ training loss, we get all examples in the span of the source training examples correct: for all $x \in \text{span}(X_{\text{src}})$ we have: $\hat{v}_{\text{src}}^\top B_{\text{fl}}^\infty x = v_*^\top B_* x$.

For all $x \in X_{\text{src}}^\perp$, from Lemma A.3 in Kumar et al. (2022a) we have that $B_{\text{fl}}^\infty x = B_{\text{ft}}(0)x = \hat{B}_{\text{src}} x$. But this means that $\hat{v}_{\text{src}}^\top B_{\text{fl}}^\infty x = \hat{v}_{\text{src}}^\top \hat{B}_{\text{src}} x$. Since we assumed that we pretrained to get $0$ loss on the source we have: $\hat{v}_{\text{src}}^\top \hat{B}_{\text{src}} x = v_*^\top B_* x$.

Combining these two cases, we get all examples in the support of $P_{\text{src}}$ correct.

$P_{\text{trg}}$ **is a mixture of** $P_{\text{src}}$ **and** $P_{\text{orth}}$. Since we get $0$ loss on the support of $P_{\text{src}}$ and support of $P_{\text{orth}}$, and $P_{\text{trg}}$ is a mixture of the two, we get $0$ loss on $P_{\text{trg}}$ as well:

$$L_{\text{trg}}(\hat{v}_{\text{src}}, B_{\text{fl}}^\infty) = 0 \tag{12}$$

$\square$

**Lemma 2.** *Suppose the representation dimension is 1-dimensional ($k = 1$) and $d_{\text{src}} > n$. Then with probability at least $1$, full fine-tuning gets positive (non-zero) loss at all times $t$:*

$$L_{\text{trg}}(v_{\text{ft}}(t), B_{\text{ft}}(t)) > 0 \tag{13}$$

*Proof.* First, we note that since $n < d_{\text{src}}$, and we only have $n$ training examples, our training examples do not span the source distribution. That is, there exists some source example $x_s \in \text{colspace}(S_{\text{src}})$ which is in the support of $P_{\text{src}}$, $x_s \neq 0$, which is orthogonal to all the training examples: $x_s \perp x_{\text{trg}}^{(i)}$ for all $1 \leq i \leq n$. Choose such an $x_s$.

We note that since $k = 1$ (the representation dimension is 1), $v_{\text{ft}}(t) \in \mathbb{R}$ is a scalar, and $B_{\text{ft}}(t), B_* \in \mathbb{R}^{1 \times d}$ are row vectors. For notational convenience, let $b(t) = B_{\text{ft}}(t)^\top$, $b_* = B_*^\top$, and $\hat{b}_{\text{src}} = \hat{B}_{\text{src}}^\top$, so we have for example $y_{\text{trg}}^{(i)} = v_* b_*^\top x_{\text{trg}}^{(i)}$.

$v_{\text{ft}}(t)$ **cannot change for zero loss.** First, we show that if $v_{\text{ft}}(t) \neq \hat{v}_{\text{src}}$ then $L_{\text{trg}}(v_{\text{ft}}(t), B_{\text{ft}}(t)) > 0$. Since $x_s$ is orthogonal to the training examples, from Lemma A.3 in Kumar et al. (2022a), we have $b(t)^\top x_s = \hat{b}_{\text{src}}^\top x_s$. Since pretraining gave us $0$ loss on the source distribution, we have $\hat{v}_{\text{src}} \hat{b}_{\text{src}}^\top x_s = v_* b_*^\top x_s$. Recall that we assumed that $w_*^\top x_s \neq 0$ if $x_s \neq 0$, which implies that $\hat{b}_{\text{src}}^\top x_s \neq 0$ since pretraining gets all the source examples right, and the ground truth label for all non-zero source examples is non-zero. But then if $v_{\text{ft}}(t) \neq \hat{v}_{\text{src}}$, we have: $v_{\text{ft}}(t) b(t)^\top x_s \neq \hat{v}_{\text{src}} \hat{b}_{\text{src}}^\top x = v_* b_*^\top x_s$. Since $P_{\text{src}}$ has density, we can construct a small ball $B$ of non-zero probability around $x_s$ such that for all $x \in B$, $v_{\text{ft}}(t) b(t)^\top x \neq v_* b_*^\top x$. This implies that $L_{\text{trg}}(v_{\text{ft}}(t), B_{\text{ft}}(t)) > 0$.

$v_{\text{ft}}(t)$ **must change for zero loss.** Next, suppose that $L_{\text{trg}}(v_{\text{ft}}(t), B_{\text{ft}}(t)) = 0$. We will show $v_{\text{ft}}(t) \neq \hat{v}_{\text{src}}$. Suppose for the sake of contradiction that $v_{\text{ft}}(t) = \hat{v}_{\text{src}}$.

From Lemma A.4 in Kumar et al. (2022a) (also see Theorem 2.2 in Du et al. (2018)), we have:

$$\hat{v}_{\text{src}}^2 - \hat{b}_{\text{src}}^\top \hat{b}_{\text{src}} = v_{\text{ft}}(t)^2 - b(t)^\top b(t). \tag{14}$$

Since $\hat{v}_{\text{src}} = v_{\text{ft}}(t)$, this gives us:

$$\hat{b}_{\text{src}}^{\top} \hat{b}_{\text{src}} = b(t)^{\top} b(t). \tag{15}$$

Let $R = \text{colspace}(S_{\text{src}})$ be the source subspace. Since $P_{\text{src}}$ is a subset of $P_{\text{trg}}$, we have that $(v_{\text{ft}}(t), b(t))$ also gets 0 loss on the source distribution, and we have:

$$\Pi_R(v_* b_*) = \Pi_R(\hat{v}_{\text{src}} \hat{b}_{\text{src}}) = \Pi_R(v_{\text{ft}}(t) b(t)). \tag{16}$$

Since $v_0 = v_{\text{ft}}(t)$, , we have:

$$\Pi_R(v_{\text{ft}}(t) \hat{b}_{\text{src}}) = \Pi_R(v_{\text{ft}}(t) b(t)). \tag{17}$$

Since $v_{\text{ft}}(t) \neq 0$ (otherwise we would get source examples wrong):

$$\Pi_R(\hat{b}_{\text{src}}) = \Pi_R(b(t)). \tag{18}$$

Let $T = \text{colspace}(S_{\text{orth}})$ be the orthogonal subspace. From Equation 15 and Equation 18, we have:

$$\|\Pi_T(\hat{b}_{\text{src}})\|_2^2 = \|\Pi_T(b(t))\|_2^2 \tag{19}$$

But to get 0 loss on $T$, we have: $\Pi_T(v_{\text{ft}}(t) b(t)) = \Pi_T(v_* b_*)$. Which implies:

$$\|\Pi_T(b(t))\|_2^2 = \frac{v_*^2}{(v_{\text{ft}}(t))^2} \|\Pi_T(b_*)\|_2^2. \tag{20}$$

From Equation 19 and since $\hat{v}_{\text{src}} = v_{\text{ft}}(t)$, we have:

$$\|\Pi_T(\hat{b}_{\text{src}})\|_2^2 = \frac{v_*^2}{(v_{\text{ft}}(t))^2} \|\Pi_T(b_*)\|_2^2. \tag{21}$$

Recall that the way we got $\hat{b}_{\text{src}}$ was we initialized $b_0 = B_0^{\top} \sim N(0, \sigma_B^2 I_d)$. We then ran gradient descent on the source distribution However, from Lemma A.3 in Kumar et al. (2022a) this does not change the projection onto components orthogonal to the source distribution. In other words, $\|\Pi_T(\hat{b}_{\text{src}})\|_2^2 = \|\Pi_T(b_0)\|_2^2$. However, this is a random variable with density, so the probability that this is exactly equal to the RHS of Equation 21 which is a fixed number, is 0. This is a contradiction.

**Wrap up.** Either ways, if $v_{\text{ft}}(t) = \hat{v}_{\text{src}}$ or $v_{\text{ft}}(t) \neq \hat{v}_{\text{src}}$ we have: $L_{\text{trg}}(v_{\text{ft}}(t), B_{\text{ft}}(t)) > 0$. □

*Proof of Theorem 1.* The proof follows directly because Lemma 2 shows that full fine-tuning gets positive (non-zero) loss but Lemma 1 gets zero loss. □

### B.3 Additional Dataset Details

Below, we provide additional information on the datasets used in our experiments.

- **CIFAR-10 → CIFAR-10-C** (Krizhevsky et al., 2009; Hendrycks & Dietterich, 2019): The task is to classify images into 10 classes, where the target distribution contains severely corrupted images. We run experiments over 14 of the corruptions (frost, gaussian blur, gaussian noise, glass blur, impulse noise, jpeg compression, motion blur, pixelate, saturate, shot noise, snow, spatter, speckle noise, and zoom blur). For the main experiments, we tune on 1000 images from CIFAR-10-C and evaluate on corrupted images from each of the corruptions. We use the data loading code from (Croce et al., 2020), which has 5 levels of severity, and we evaluate with the most severe level. In our main experiments, we report the accuracies averaged across all corruptions and the average std error for all corruptions.

- **ImageNet → ImageNet-C** (Deng et al., 2009; Hendrycks & Dietterich, 2019): The task is to classify images into 1000 classes, where the target distribution contains severely corrupted images. We run experiments over 15 of the corruptions (brightness, contrast, defocus blur, elastic transform, fog, frost, Gaussian noise, glass blur, impulse noise, jpeg compression, motion blur, pixelate, shot noise, snow, zoom blur). For the main experiments, we tune on 5000 images from ImageNet-C, evenly split between classes, giving 5 corrupted images per class and evaluate on corrupted images from each of the corruptions. Similar to CIFAR-10-C, we evaluate with the most severe level. We also report the accuracies averaged across all corruptions and the average std error for all corruptions.

- **Living-17** and **Entity-30** (Santurkar et al., 2020): The task is to classify images into one of 17 animal categories or one of 30 entities. These datasets present subpopulation shifts, in that while the ID and OOD distributions have the same overall classes, they contain different subpopulations of those classes. For Living-17, we tune on 850 images from the target distribution, evenly split between the 17 classes, giving 50 images per class. For Entity-30, we tune on 1500 images from the target distribution, evenly split between the 30 classes, giving 50 images per class.

- **Waterbirds** (Sagawa et al., 2019): The task is to classify images as being a "waterbird" or "landbird". The label is spuriously correlated with the image background, which is either "land" or "water." The source distribution is the training set while the target distribution is a balanced subset with equal amounts of each bird on each background. In the training data, 95 % of the waterbirds appear on water backgrounds, and 95% of the landbirds appear on land backgrounds, so the minority groups contain far fewer examples than the majority groups. We tune on 400 images from the target distribution, evenly split between the 4 groups of (bird, background) pairs, giving 100 images per group.

- **CelebA** (Sagawa et al., 2019): The task is to classify the hair color in images as "blond" or "not blond", and the label is spuriously correlated with the Male attribute. The source distribution is the training set while the target distribution is a balanced subset with equal amounts of each of the four (hair color, gender) groups. We tune on 400 images from the target distribution, evenly split between the 4 groups of (hair color, gender) pairs, giving 100 images per group.

- **Camelyon17** (Bandi et al., 2018): This dataset is part of the WILDS (Koh et al., 2021) datasets and contains roughly 450,000 images in the source distribution (Train) and 84,000 images in the target distribution (OOD test) of size $96 \times 96$. It comprises of medical images collected from 5 hospitals where difference in devices/data-processing between different hospitals produces a natural distribution shift. We pre-train on the 450,000 images of the source distribution and use 100 label-balanced images (50 per class) from the target distribution for fine-tuning. We use another 100 label-balanced target distribution images for tuning hyper-parameters, and report the performance of the fine-tuned model on the rest of the images of the target distribution.

- **FMoW** (Christie et al., 2018): This is also part of the WILDS (Koh et al., 2021) datasets and its source distribution contains 520,000 satellite images of size $224x224$ from 5 geographic regions. The task is to classify one of 62 building or land use types. For target distribution, we use Africa test, i.e., the subset of the OOD test data belonging to Africa region, which has roughly 2,500 images. We use 62 label-balanced images from target distribution for fine-tuning and report the accuracy on the rest of the target distribution images.

## B.4 Additional Details for Supervised Transfer Learning Experiments

Below, we provide additional details for our experiments on real data, including tuning details. For all datasets and experiments, we **early stop** according to the best accuracy on a held-out validation subset of the labeled target data.

- **CIFAR-10 $\rightarrow$ CIFAR-10-C** (Krizhevsky et al., 2009; Hendrycks & Dietterich, 2019) and CIFAR-Flip: We use the Standard pre-trained model from (Croce et al., 2020), which is trained on the source CIFAR-10 distribution. We fine-tune on the labeled target data for 15 total epochs. We tune over the 3 learning rates {1e-3, 1e-4, 1e-5} for all methods except last-layer fine-tuning, where we tune over {1e-1, 1e-2, 1e-3}, and we use a weight decay of 0.0001 for all methods.

- **ImageNet $\rightarrow$ ImageNet-C** (Deng et al., 2009; Hendrycks & Dietterich, 2019): We use the Standard pre-trained model from (Croce et al., 2020), which is trained on the source ImageNet distribution. We then fine-tune on the labeled target data for 10 total epochs. We tune over the 3 learning rates {1e-3, 1e-4, 1e-5} for all methods, and we use a weight decay of 0.0001 for all methods.

- **Living-17** and **Entity-30** (Santurkar et al., 2020): We first train on the source data for 5 epochs, tuning only the head for 3 epochs and then fine-tuning all for 2 more epochs, following LP-FT (Kumar et al., 2022a) and using the Adam optimizer. We then fine-tune on the labeled target data for 15 epochs. We tune over the 3 learning rates {0.0005, 0.0001, 0.00001} for all methods and do not use any weight decay.

- **Waterbirds** (Sagawa et al., 2019): We first start with a ResNet-50 pretrained on ImageNet and train on the source distribution for 300 epochs, taking the best checkpoint based on early stopping and using the Adam optimizer. We then fine-tune on the labeled target data for 100 total epochs. We tune over the 3 learning rates {0.005, 0.001, 0.0005} for all methods and use a weight decay of 0.0001.

| Method | mCE (%) |
|---|---|
| Vanilla ResNet-50 (Hendrycks & Dietterich, 2019) | 76.7 |
| Cross-Val | **61.7** |
| Full Fine-tuning | 62.8 |
| Gradual (First → Last) | 62.4 |
| Gradual (Last → First) | 63.6 |
| L1 Regularize (Xuhong et al., 2018) | 64.7 |
| Auto-SNR | 62.9 |
| Auto-RGN | **61.9** |

Table 6: For completeness, we report the mean corruption error (mCE) on ImageNet-C, which weights the target distribution error by the difficulty of the corruption. We report the average accuracies on the target distribution in Figure 2 and Table 4, and we see that the two metrics are correlated for our experiments, as the best performing methods according to average accuracy are also the best for mCE.

| First Layers | Block 1 | Block 2 | Block 3 | Block 4 | Last | All | No Tuning |
|---|---|---|---|---|---|---|---|
| 74.6 (1.9) | 75.3 (1.9) | 81.6 (1.2) | **86.1 (0.7)** | 82.9 (1.5) | 72.6 (1.3) | 77.3 (0.4) | 65.6 (0) |

Table 7: Surgical fine-tuning results on Living-17 (% accuracy) initialized with a CLIP ViT-B/16. We find that similar to the results using a ResNet-50 architecture, fine-tuning a single parameter block with this vision transformer architecture outperforms full fine-tuning, and in particular, a middle block still performs best for this feature-level shift.

- **CelebA** (Sagawa et al., 2019): We first start with a ResNet-50 pretrained on ImageNet and train on the source distribution for 50 epochs, taking the best checkpoint based on early stopping and using the Adam optimizer. We then fine-tune on the labeled target data for 50 total epochs. We tune over the 3 learning rates {0.001, 0.0005, 0.0001} for all methods and use a weight decay of 0.0001.

- **Camelyon17** (Bandi et al., 2018; Koh et al., 2021): We start with a vision transformer, CLIP ViT-B/16 (Radford et al., 2021), pre-trained on the CLIP datasets. Next we fine-tune the model on Camelyon17 train dataset using an SGD optimizer with initial learning rate 0.0001 for 3 epochs. We use a cosine annealing learning rate scheduler (Loshchilov & Hutter, 2017) and batch size 32. Finally, we fine-tune on the labeled target data for 10 total epochs with the same setting as before, except we tune over learning rates $\{10^{-4}, 10^{-5}, 3 \times 10^{-5}, 7 \times 10^{-5}, 10^{-6}, 3 \times 10^{-6}, 10^{-7}, 10^{-8}\}$

- **FMoW** (Christie et al., 2018; Koh et al., 2021): Similar to Camelyon17, we start with a vision transformer, CLIP ViT-B/16 (Radford et al., 2021), pre-trained on the CLIP datasets. Next we fine-tune the model on FMoW train dataset using an SGD optimizer with initial learning rate 0.0003 for 5 epochs. We use a cosine annealing learning rate scheduler (Loshchilov & Hutter, 2017) and batch size 32. Finally, we fine-tune on the labeled target data for 10 total epochs with the same setting as before, except we tune over learning rates $\{10^{-6}, 10^{-5}, 0.0003, 0.0001, 0.001, 0.01, 0.1, 0.25\}$.

In Table 6, for completeness, we additionally report the mean corruption error (mCE), which weights the target distribution error by the difficulty of the corruption, on ImageNet-C since that is a common metric used for the dataset. We find that these results give similar conclusions as the average accuracies reported in Table 4, with Cross-Val and Auto-RGN performing the best.

We additionally include an ablation where we use a CLIP ViT-B/16 (Vision Transformer) as our initial model pretrained on the WebImageText dataset. This model consists of 2 first layers followed by 4 transformer blocks and 2 last layers. We analyze surgical fine-tuning with this model architecture on the Living-17 dataset, which has a feature-level shift. In Table 7, we find that our results in Section 2 hold similarly using this vision transformer, as tuning only a middle block outperforms full fine-tuning or tuning any other block of layers.

### B.5 MORE LARGE VISION TRANSFORMER EXPERIMENTS

Prior work has shown that when fine-tuning vision transformers, task accuracy on held-out data is substantially higher when using the AdamW optimizer rather than SGD (Kumar et al., 2022b). We evaluate the performance of surgical fine-tuning on two large pre-trained vision transformer models (CLIP ViT-B/16 and ViT-L/14) while fine-tuning with AdamW. We follow the experimental setting of Kumar et al. (2022b) as closely as possible:

| Dataset
Architecture
Optimizer | Camelyon17
ViT-B/16
SGD | Camelyon17
ViT-B/16
AdamW | FMoW
ViT-B/16
SGD | FMoW
ViT-B/16
AdamW | FMoW
ViT-L/14
AdamW |
|---|---|---|---|---|---|
| No fine-tuning | 86.2 | 96.2 | 35.5 | 41.7 | 50.3 |
| All | 92.3 (1.7) | 96.4 (0.4) | 38.9 (0.5) | 29.2 (2.6) | **52.9 (1.2)** |
| Embedding | **95.6 (0.4)** | **96.9 (0.1)** | 36.0 (0.1) | 41.8 (0.1) | 50.1 (1.6) |
| First three blocks | 92.5 (0.5) | 95.0 (0.2) | 39.8 (1.0) | 26.7 (1.3) | 51.6 (0.9) |
| Last three blocks | 87.5 (4.1) | 96.6 (0.1) | **44.9 (2.6)** | **46.1 (2.3)** | **52.4 (0.6)** |
| Last layer | 90.1 (1.5) | 96.7 (0.1) | 36.9 (5.5) | 42.6 (0.1) | **52.3 (0.3)** |

Table 8: OOD set accuracies after surgically fine-tuning different parameters in vision transformer models for two WILDS datasets. Bold numbers represent superior results for a dataset, and we also report the standard deviation from runs with 3 different seeds.

- **Camelyon17** (Bandi et al., 2018; Koh et al., 2021). We first train a pre-trained CLIP ViT-B/16 (Radford et al., 2021) model on the train split of the Camelyon17 dataset using an AdamW optimizer with initial learning rate $10^{-6}$ for 3 epochs. We use a cosine annealing learning rate scheduler (Loshchilov & Hutter, 2017) and batch size 32. Finally, we fine-tune on the labeled target data for 10 total epochs with the same setting as before, except we tune over learning rates $\{10^{-4}, 10^{-5}, 3 \times 10^{-5}, 7 \times 10^{-5}, 10^{-6}, 3 \times 10^{-6}, 10^{-7}, 10^{-8}\}$.

- **FMoW** (Christie et al., 2018; Koh et al., 2021). We similarly train a pre-trained CLIP ViT-B/16 or ViT-L/14 model on the train split of the FMoW dataset with initial learning rate $10^{-5}$ for 5 epochs. We use a cosine annealing learning rate scheduler (Loshchilov & Hutter, 2017) and batch size 32. For the ViT-L/14 setting, we use larger $336 \times 336$ images. Finally, we fine-tune on labeled target data for 10 total with the same setting as before, tuning over learning rates $\{10^{-4}, 10^{-5}, 3 \times 10^{-5}, 7 \times 10^{-5}, 10^{-6}, 3 \times 10^{-6}, 10^{-7}, 10^{-8}\}$.

We show AdamW fine-tuning results in Table 8. While surgically fine-tuning the right layer continues to improve over no fine-tuning, the relative advantage compared to fine-tuning all layers is smaller than what we observed with SGD in Table 7. We observe instability in fine-tuning all layers of a ViT-B/16 network for FMoW target distributions. Fine-tuning later layers seems to consistently improve performance without running into such instability issues. We leave further investigation of such properties of fine-tuning ViT models to future work.

### B.6 Complete unsupervised adaptation results

**Method.** We experiment with MEMO (Zhang et al., 2021a) as our unsupervised adaptation method. Given a test image $x$, MEMO first takes an "adapt" stage, where it minimizes the marginal entropy over standard augmentations of $x$, then it takes a "test" step, where the network predicts a label for $x$. Note that MEMO tests a single image at a time, i.e., the test batch size is 1. We also consider the two following variations.

- **Episodic**: This version is discussed in the origin work. Here after predicting the labels, we reset the weights of the network to the pre-trained ones, i.e., we undo the "adapt" stage.

- **Online**: We also consider the online variation of MEMO, where we do not reset the weights after each test image, i.e., we accumulate the "adaptation" changes over test images.

We have also experimented with TENT (Wang et al., 2020), but since TENT only updates the batchnorm modules (whereas MEMO updates all parameters), freezing parameters with TENT did not produce expected results and we did not pursue it further.

**Dataset and Network.** We use the CIFAR-10-C and ImageNet-C corruption datasets for our experiments. For CIFAR-10-C, we use the same ResNet-26 (He et al., 2015) pre-trained model used by MEMO (Zhang et al., 2021a), which is available in their GitHub repository. For ImageNet-C, we use RVT*-small architecture and pre-trained weights used by Zhang et al. (2021a).

**Hyper-parameters.** For CIFAR-10-C, we use 1000 corrupted test image for hyper-parameter tuning and report the test accuracy on the held out 9000 examples. We consider the following hyper-parameter grid:

| Settings | Tuned Layers | gauss | impulse | shot | fog | frost | snow | elast | brit | contr | defoc |
|----------|--------------|-------|---------|------|-----|-------|------|-------|------|-------|-------|
| | | | | | | | | Corruption | | | |
| No Adaptation | | 51.6 | 49.7 | 55.2 | 72.0 | 66.9 | 75.9 | 74.4 | 85.9 | 70.3 | 75.9 |
| Episodic | All | 56.31 | 56.39 | 59.94 | 77.42 | 71.93 | 78.63 | **78.99** | **88.15** | 72.66 | 76.32 |
| | First layer | 53.11 | 51.29 | 56.53 | 73.27 | 68.41 | 77.16 | 76.46 | 86.53 | 70.82 | 76.11 |
| | First 2 layers | 53.20 | 51.21 | 56.58 | 73.48 | 68.64 | 77.06 | 76.41 | 86.47 | 70.67 | 76.15 |
| | First 2 blocks | 53.20 | 51.36 | 56.63 | 73.37 | 68.61 | 77.22 | 76.45 | 86.56 | 70.73 | 76.09 |
| | Last | 51.69 | 49.82 | 55.24 | 72.01 | 67.05 | 75.91 | 74.56 | 85.98 | 70.28 | 75.98 |
| Online | All | 51.48 | 49.53 | 55.85 | 75.69 | 72.64 | 77.36 | 76.16 | 86.83 | 70.88 | 79.89 |
| | First layer | **68.14** | 59.21 | 71.10 | **79.61** | **76.20** | 79.29 | 76.86 | 86.72 | 74.26 | **82.18** |
| | First 2 layers | 68.00 | **60.92** | **73.07** | 79.22 | 76.16 | 79.34 | 76.18 | 86.65 | 73.97 | 81.97 |
| | First 2 blocks | 67.84 | 59.30 | 70.96 | 79.11 | 75.91 | **79.44** | 76.46 | 86.91 | **74.67** | 82.12 |
| | Last | 51.61 | 49.71 | 55.21 | 71.99 | 66.92 | 75.87 | 74.49 | 86.00 | 70.27 | 75.91 |

Table 9: MEMO (Zhang et al., 2021a) with parameter freezing results on CIFAR-10-C on 10 representative corruptions and severity level 5. Bold numbers represent superior results for a particular corruption.

- Learning rate: $10^{-3}$, $10^{-4}$, $10^{-5}$ and $10^{-6}$, then $2.5x$, $5x$, $0.5x$ of the best learning rate from before.
- Steps: 1, 2.
- Weight decay: 0, $10^{-3}$, $10^{-2}$, $10^{-1}$.
- Number of augmentations per image: 32

For ImageNet-C, we do not do any hyper-parameter tuning and simply use the best hyper-parameters described by Zhang et al. (2021a), which are:

- Learning rate: 0.00001
- Weight decay: 0.01
- Steps: 1
- Number of augmentations per image: 32 (This is 64 in Zhang et al. (2021a), but we use 32 for computational cost)

Moreover, for ImageNet-C, the experiments are done over 2000 test images (the first 2 test image per class for each of the 1000 classes) instead of the entire test set of $50,000$ images, for computational reasons. This produces numbers that are slightly different from the original work, hence we include the no adaptation baselines as well for fair comparison.

Finally, in practice, we saw that AdamW and SGD optimizers work better for the episodic and online setting, respectively.

**Layers.** We use the following naming convention for the layers of ResNet-26:

- **First**: Only the first *conv* layer.
- **First 2 layers**: The first *conv* layer of the entire network, and the first *conv* layer within the first block.
- **First 2 blocks**: The first *conv* layer, and the first block.
- **Last**: The last fully-connected (FC) layer.

For RVT*-small:

- **First layer**: First conv layer inside the first transformer block.
- **First block**: First transformer block.
- **Last**: Head or the final fully connected layer.

**Results.** Table 9 and Table 10 show results for MEMO with parameter freezing for CIFAR-10-C and ImageNet-C respectively on severity level 5 and on a representative set of corruptions.

| Settings | Tuned Layers | Corruption | | | | | | | | | | | | | |
|---|---|---|---|---|---|---|---|---|---|---|---|---|---|---|---|
| | | gauss | impul | shot | fog | frost | snow | brit | contr | elast | glass | motion | zoom | pixel | jpeg |
| No Adapt | | 43.80 | 45.90 | 42.40 | 55.20 | 45.95 | 50.05 | 73.95 | 53.25 | 35.00 | 19.35 | 40.35 | 32.05 | 52.60 | 60.60 |
| Episodic | All | 45.35 | 47.40 | 44.60 | 55.35 | 47.05 | 50.25 | 74.35 | **54.85** | 36.25 | 20.55 | 41.40 | 33.95 | 55.55 | **61.70** |
| | First layer | 44.00 | 46.00 | 42.40 | 55.20 | 45.90 | 50.10 | 74.0 | 53.30 | 35.20 | 19.45 | 40.60 | 32.15 | 52.80 | 60.60 |
| | First block | 44.10 | 46.30 | 42.60 | 55.40 | 46.00 | 50.15 | 74.10 | 53.20 | 35.15 | 19.75 | 40.75 | 32.25 | 53.20 | 60.75 |
| | Last | 43.75 | 45.95 | 42.40 | 55.25 | 46.0 | 50.05 | 73.95 | 53.30 | 35.05 | 19.35 | 40.40 | 32.05 | 52.60 | 60.60 |
| Online | All | 1.05 | 0.90 | 1.40 | 0.35 | 3.40 | 1.60 | 0.80 | 0.80 | 2.75 | 1.05 | 2.50 | 2.25 | 2.70 | 3.40 |
| | First layer | 44.70 | 46.30 | 43.40 | 46.50 | 47.0 | 50.9 | 74.15 | 48.85 | 36.40 | 20.20 | **41.65** | 32.50 | 55.70 | 61.05 |
| | First block | **46.50** | **49.65** | **46.75** | **55.80** | **47.40** | **52.55** | **74.80** | 54.15 | **40.50** | **24.75** | 41.45 | **34.15** | **56.85** | 61.05 |
| | Last | 43.80 | 45.90 | 42.40 | 55.15 | 45.95 | 50.00 | 73.90 | 53.20 | 35.05 | 19.35 | 40.35 | 32.05 | 52.60 | 60.60 |

Table 10: MEMO (Zhang et al., 2021a) with parameter freezing results on ImageNet-C on 14 representative corruptions and severity level 5. Bold numbers represent superior results for a particular corruption.

