# OpenReview forum: "Surgical Fine-Tuning Improves Adaptation to Distribution Shifts"
_ICLR.cc/2023/Conference — ICLR 2023 poster_

### Official Review · Reviewer_Y4ag · 2022-10-23

**Confidence:** 3
**Correctness:** 3
**Technical Novelty And Significance:** 3
**Empirical Novelty And Significance:** 3
**Recommendation:** 8

**Clarity, Quality, Novelty And Reproducibility:**

## Clarity and Quality
- Presentation is good and the paper is easy to follow.
## Novelty
- This paper present a counter-intuitive finding, which is novel.
## Reproducibility
- Opensource is not guaranteed in the paper. Experiment details are in the appendix. It should not be difficult to reproduce the paper.


**Strength And Weaknesses:**

## Strength
- This paper present a counter-intuitive finding, with detailed experimental and theoretical analysis
## Weakness
- Datasets used are relatively small. It is fine that this work focus on "fine-tune the pre-trained model on a small target dataset". However, it would be more persuasive to show some results on large model + mid-size dataset. In industry, this is a more common setting.

**Summary Of The Paper:**

In this paper, an interesting finds are demonstrated: instead of always finetuning the last layer of the model, tuning different layers works best for different types of distribution shifts. 3 types of shifts are discussed: (1) Input-level shift, first-layer finetuning is better; (2) Feature-level shift, mid-later blocks are better; (3) Output-level shift, last layer is better;

**Summary Of The Review:**

This paper present a counter-intuitive finding, with detailed experimental and theoretical analysis. However, the major concern is that the finding is validated on small datasets, which might not be able to generalize to larger datasets finetuning.

---

> ### Author Response · Authors · 2022-11-13
> **Response to Reviewer Y4ag**
>
> We thank you for your review and positive assessment of our work. If our response does not address all of your concerns, please let us know!
>
> > Datasets used are relatively small. It is fine that this work focus on "fine-tune the pre-trained model on a small target dataset". However, it would be more persuasive to show some results on large model + mid-size dataset. In industry, this is a more common setting.
>
> We evaluated surgical fine-tuning on two large real-world datasets (Camelyon-WILDS, FMoW-WILDS), which have more severe distribution shifts than the datasets considered in the original submission. The Camelyon and FMoW datasets consist of tumor and satellite images, where different distributions correspond to different hospitals and regions. These datasets are quite large-scale: Camelyon consists of 450,000 images of size 96 x 96 from 5 hospitals, and FMoW consists of 520,000 images of size 224 x 224 from 5 regions. For this experiment, we use CLIP ViT-B/16, a large pretrained vision Transformer model. This setting can be considered a “large model + mid-size dataset” as per your suggestion. We summarize the results below:
>
>
> | Layers | Camelyon-WILDS | FMoW-WILDS |
> | ----------- | ----------- | ----------- |
> | No Adaptation | 86.2 | 35.5 |
> | Full Fine-tuning | 92.3 (1.7) | 38.9 (0.5) |
> | Embedding Layer  | **95.6 (0.4)** | 36.0 (0.1) |
> | First 3 attention blocks  | 92.5 (0.5) | 39.8 (1.0) |
> | Last 3 attention blocks | 87.5 (4.1) | **44.9 (2.6)** |
> | Last layer | 90.1 (1.5) | 36.9 (5.5) |
>
> Even in large-scale distribution shift conditions where many shift types (input, feature, output) may appear together, applying surgical fine-tuning to a suitable layer subset achieves substantially higher test set accuracy. Of the specific shift types considered in the paper, Camelyon17 is most close to an input-level shift because the most salient shift between images from different hospitals is the color shift due to lighting differences. On the other hand, the change between different regions in FMoW can be seen as close to a feature-level shift because building shape and spacing are most salient in satellite imagery. The results above agree with this intuition: tuning the earliest embedding layer works best for Camelyon17, and tuning later attention blocks works best for FMoW. We have added these results to Table 1 in the paper.
>
> > Opensource is not guaranteed in the paper.
>
> In the final version, we will include a link to a public repository as a footnote in page 1.

---

> > ### Comment · Reviewer_Y4ag · 2022-11-13
> > **Thanks for the response**
> >
> > Thanks for the response and sorry for missing the foot note :). The results on large model + real world datasets are aligned with the claims in the paper, I would be happy to raise the rating.

---

> > > ### Author Response · Authors · 2022-11-16
> > > **Checking in**
> > >
> > > Thank you again for your review and quick response to our rebuttal. We noticed that your rating is still at its original value (6) and wanted to check in on this. Please let us know if you have any further questions!

---

> > > > ### Comment · Reviewer_Y4ag · 2022-11-16
> > > > **updated**
> > > >
> > > > Updated, not sure if this is the right way to do so, for other conferences, there will be a final rating options different from the original one

---

### Official Review · Reviewer_QAba · 2022-10-24

**Confidence:** 3
**Correctness:** 3
**Technical Novelty And Significance:** 3
**Empirical Novelty And Significance:** 3
**Recommendation:** 8

**Clarity, Quality, Novelty And Reproducibility:**

They leveraged their empirical findings with verification of theoretical equations to propose a new training algorithm that outperforms a conventional approach. Their work is interesting to read and novel as most prior works have shown just experimental results without theoretical verification and automation of selection of layers to tune for transfer learning. The reviewer thinks most of results shown in the paper can be reproduced.

**Strength And Weaknesses:**

Strength
• For empirical studies, they categorized distributional shifts into three main scenarios including input, feature, and output levels to analyze optimal selection of layers to tune for each scenario with plausible explanations.
• This type of study usually reports experimental results only without theoretical verification. But this paper somehow provides theoretical verification using a simplified two-layer neural network.
• Diverse experiments have been conducted to make their claims.
• Beyond the explanations of the empirical results, they leveraged the findings to develop a training pipeline with automatically selecting layers to tune, and demonstrated the proposed approach achieves better performance than simply full fine-tuning which has been mostly used by researchers without any doubts.
• In addition to supervised learning, they analyzed unsupervised learning that otherwise would be requested by reviewers.
Weakness
• Can you explain the rationale about why only contagious subset of layers need to be tuned instead of discrete subset of layers?
• In this study, distribution shifts are categorized into three levels including input, feature, and outputs. However, it would be interesting to show results for combination of scenarios like input+feature, feature+outputs, inputs+outputs, and input+feature+outputs shifts. Especially, when we want to fine-tune ImageNet pretrained models on medical images (which is commonly used by biomedical researchers), do we expect full fine-tuning achieves the best performance instead of subset of layers to tune as natural images to medical images have distributional shifts of input, feature, and outputs levels. These results could be much more interesting because large distributional shift scenarios are more like real word situation compared to what’s shown in this paper.

**Summary Of The Paper:**

This paper shows fine-tuning a (contiguous) subset of layers matches or outperforms commonly used fine-tuning approaches including full fine-tuning or fine-tuning the last few layers. The authors evaluated the performance of surgical fine-tuning diverse layers of choice on 7 different distribution shift scenarios (input, feature, and output-level shifts). In addition to these empirical results, they tried to validate theoretically in an idealized two-layer neural network setup to explain why the earlier and later layer fine-tuning achieve best performance for input and output-level shifts, respectively. At last, the authors investigated several criteria for automatic selection of which layers to tune for better performance, and they empirically demonstrated that the automatic fine-tuning outperforms full fine-tuning.

**Summary Of The Review:**

The authors investigated how fine-tuning can be adapted to diverse distribution shifts by showing empirical results and theoretical verification, followed by proposing a new type of regularization (several criteria to control extent of tuning per layer) to achieve a better performance than a commonly used approach (full fine-tuning). But, the paper can be more interesting to read by the potential readers if they can show results for real-world scenarios that are likely to have more than two types of distribution shifts.

---

> ### Author Response · Authors · 2022-11-13
> **Response to Reviewer QAba**
>
> We thank you for your detailed comments and appreciate your acknowledgment of many strengths of our work. We include clarifications to individual points below. If there are any remaining questions, please let us know!
>
> > ...do we expect full fine-tuning achieves the best performance instead of subset of layers to tune as natural images to medical images have distributional shifts of input, feature, and outputs levels. These results could be much more interesting because large distributional shift scenarios are more like real word situation compared to what’s shown in this paper.
>
>
> We evaluated surgical fine-tuning on two large real-world datasets (Camelyon-WILDS, FMoW-WILDS), which have more severe distribution shifts than the datasets considered in the original submission. The Camelyon and FMoW datasets consist of tumor (medical, as you suggested) and satellite images, where different distributions correspond to different hospitals and regions. Therefore, these datasets can be considered a mix of several naturally occurring distribution shifts. These datasets are quite large-scale: Camelyon consists of 450,000 images of size 96 x 96 from 5 hospitals, and FMoW consists of 520,000 images of size 224 x 224 from 5 regions. For this experiment, we use CLIP ViT-B/16, a large pretrained vision Transformer model. We summarize the results below:
>
> | Layers | Camelyon-WILDS | FMoW-WILDS |
> | ----------- | ----------- | ----------- |
> | No Adaptation | 86.2 | 35.5 |
> | Full Fine-tuning | 92.3 (1.7) | 38.9 (0.5) |
> | Embedding Layer  | **95.6 (0.4)** | 36.0 (0.1) |
> | First 3 attention blocks  | 92.5 (0.5) | 39.8 (1.0) |
> | Last 3 attention blocks | 87.5 (4.1) | **44.9 (2.6)** |
> | Last layer | 90.1 (1.5) | 36.9 (5.5) |
>
> Even in natural distribution shift conditions where many shift types (input, feature, output) may appear together, applying surgical fine-tuning to a suitable layer subset achieves substantially higher test set accuracy. Of the specific shift types considered in the paper, Camelyon17 is most close to an input-level shift because the most salient shift between images from different hospitals is the color shift due to lighting differences. On the other hand, the change between different regions in FMoW can be seen as close to a feature-level shift because building shape and spacing are most salient in satellite imagery. The results above agree with this intuition: tuning the earliest embedding layer works best for Camelyon17, and tuning later attention blocks works best for FMoW. We have added these results to Table 1 in the paper.
>
> > Can you explain the rationale about why only contagious subset of layers need to be tuned instead of discrete subset of layers?
>
> In our work, we hypothesize that distribution shifts often correspond to local changes in the data generative process, especially when the source and target distributions have a shared structure, as in the datasets we consider. We further hypothesize that neural networks store information in a depthwise modular way, where nearby layers store similar information. Therefore, it suffices to fine-tune only the network region corresponding to the change responsible for the distribution shift; our experiments demonstrate that a small contiguous subset of layers suffices for many distribution shifts. In more complicated situations where these assumptions do not hold, a contiguous block may not be best: for example, fine-tuning the first and last layers may be best for some distribution shifts. Such cases can be an exciting avenue for future work.

---

> ### Author Response · Authors · 2022-11-16
> **Checking in**
>
> Thank you again for all of your detailed comments! Please let us know if our response has addressed your concerns and if there's anything else you have questions about.

---

### Official Review · Reviewer_eWpB · 2022-10-25

**Confidence:** 4
**Correctness:** 4
**Technical Novelty And Significance:** 3
**Empirical Novelty And Significance:** 3
**Recommendation:** 6

**Clarity, Quality, Novelty And Reproducibility:**

The work is explained well and has novel aspects in it. In terms of reproducibility, I was not able to find the code in the supplementary material but it should be relatively straightforward to reproduce it

**Strength And Weaknesses:**

Strength(s):
1. The paper nicely and cleanly classifies the distribution shifts into three types and empirically shows how different fine-tuning strategies are optimal for different types of distribution shifts.
2. The synthetic experiment in the paper is clever and successfully demonstrates that surgically fine-tuning the relevant layers/blocks can result in a significant performance gain.
3. New strategies (Auto-SNR and Auto-RGN) for automatically selecting which layers to fine-tune were a nice outcome of the theoretical analysis performed by the authors.
4. The empirical observations to fine-tune only a subset of layers for different distribution shifts are well supported by multiple datasets having different types of distribution shifts.

Weakness(es)/Suggestion(s):
1. For table 3, in addition to the L1 regularization method, it would be nice to include other methods like gradually unfreezing or different learning rates for different layers, etc.

(some methods from the existing work:
A. Simon Kornblith, Jonathon Shlens, and Quoc V Le. Do better imagenet models transfer better? In Proceedings of the IEEE/CVF conference on computer vision and pattern recognition, pp. 2661–2671, 2019. [page 1]
B. Hao Li, Pratik Chaudhari, Hao Yang, Michael Lam, Avinash Ravichandran, Rahul Bhotika, and Stefano Soatto. Rethinking the hyperparameters for fine-tuning. In International Conference on Learning Representations, 2020. URL https://openreview.net/forum id=B1g8VkHFPH. [page 1, 9]
C. Zhiqiang Shen, Zechun Liu, Jie Qin, Marios Savvides, and Kwang-Ting Cheng. Partial is better than all: Revisiting fine-tuning strategy for few-shot learning. In Proceedings of the AAAI Conference on Artificial Intelligence, volume 35, pp. 9594–9602, 2021. [page 1, 9]
D. Youngmin Ro and Jin Young Choi. Autolr: Layer-wise pruning and auto-tuning of learning rates in fine-tuning of deep networks. In Proceedings of the AAAI Conference on Artificial Intelligence, volume 35, pp. 2486–2494, 2021. [page 1]
E. Miguel Romero, Yannet Interian, Timothy Solberg, and Gilmer Valdes. Targeted transfer learning to improve performance in small medical physics datasets. Medical Physics, 47(12):6246–6256, 2020. [page 1])

2.  Although the observation of fine-tuning different subsets of layers for different distribution shifts is an interesting one, it is not clear how the method of surgically fine-tuning a network would compare to existing methods for domain adaptation or methods that perform test-time adaptation (adjusting the covariate shift, etc.) [F,G,H]. It would strengthen the paper if authors can show that surgical fine-tuning is a better and simpler option to tackle distribution shifts as compared to training exotic models that enable robustness or allow for domain adaptation.

F. Improving robustness against common corruption by covariate shift adaptation, NeurIPS 2020
G. Adaptive Denoising via GainTuning, NeurIPS 2021
H. Be Like Water: Robustness to Extraneous Variables Via Adaptive Feature Normalization, 2020

3. Is the reported metric for CIFAR-C and IN-C (1- mean corruption error (mCE)), in Tables 1,2, and 3, and Figures 2, 3 and 4? Because mCE is a standard metric to track when dealing with corruption datasets.






**Summary Of The Paper:**

The paper performs an analysis of what the fine-tuning strategy should be when encountering various distribution shifts. They classify the different distribution shifts into three categories and show that for each category of distribution shift, different layers of the model should be surgically fine-tuned. This analysis results in interesting outcomes, for example, for shifts where the inputs are changing (like CIFAR-C), contrary to the conventional wisdom of fine-tuning the last few layers to re-use the learned features, it is better to fine-tune only the early layers of the network. The paper also discusses and compares some ways to automatically decide which layers to fine-tune.

**Summary Of The Review:**

Overall, although the work has some interesting analyses and outcomes/results, it is still unclear how the method of surgically fine-tuning a network would compare to existing methods for domain adaptation or methods that perform test-time adaptation (adjusting the covariate shift, etc.). Therefore, my recommendation for this paper would be marginally below the acceptance threshold (5).

---

> ### Author Response · Authors · 2022-11-13
> **Response to Reviewer eWpB**
>
> We thank you for your detailed feedback. We include clarifications and answers to individual concerns below. In light of these new clarifications, please let us know if you have any remaining concerns. We are happy to answer any further questions you may have.
>
> > For table 3, in addition to the L1 regularization method, it would be nice to include other methods like gradually unfreezing or different learning rates for different layers, etc.
>
> We added two variations of gradually unfreezing layers as additional comparisons in Table 3: Gradual Unfreeze (First -> Last) and Gradual Unfreeze (Last -> First). In the former, all layers except the first block are frozen initially, and the other layers are sequentially unfrozen at evenly spaced out epochs with the same learning rate. The latter is in the opposite order and is similar to previously proposed fine-tuning strategies ([1], [2], [3]), where all layers except the last layer are frozen, and earlier layers are gradually unfrozen until the entire model is being tuned.
> Cross-Val and Auto-RGN consistently outperform these two approaches across our 7 real-data tasks. Still, the directionality of the Gradual Unfreeze results is consistent with that of surgical fine-tuning: unfreezing earlier leads to better performance in input-level shifts, and unfreezing later layers is better in output-level shifts. The table in the paper has been updated to include the new results below:
>
> | Method      | CIFAR-C | IN-C | Living-17 | Entity-30 | CIFAR-F | Waterbirds | CelebA | Rank |
> | ----------- | ----------- | ----------- | ----------- | ----------- | ----------- | ----------- | ----------- | ----------- |
> | No Adaptation | 52.6 | 18.2 | 80.7 | 58.6 | 0.0 | 31.7 | 27.8 | - |
> | Cross-Val (oracle) | 82.8 (0.6) | 51.6 (0.1) | 93.2 (0.3) | 81.2 (0.6) | 93.8 (0.1) | 89.9 (1.2) | 86.2 (0.8) | - |
> | Full Fine-Tuning | 79.9 (0.7) | 50.7 (0.1) | 92.8 (0.7) | 79.3 (0.6) | 85.9 (0.4) | **88.0 (1.2)** | 82.2 (1.3) | 2.71 |
> | Gradual Unfreeze (First->Last) | 80.5 (0.7) | 50.7 (0.1) | 90.1 (0.1) | 69.9 (3.0) | 81.9 (0.6) | 87.5 (0.2) | 71.9 (0.7) | 4.71 |
> | Gradual Unfreeze (Last->First) | 78.8 (0.8) | 49.8 (0.2) | 90.6 (0.2) | 77.8 (0.5) | 84.3 (0.9) | 87.8 (0.1) | 78.5 (2.9) | 4.0 |
> | L1 Regularize | 81.7 (0.6) | 48.8 (0.3) | 93.4 (0.5) | 78.4 (0.1) | 84.2 (1.2) | 87.6 (1.9) | **82.6 (1.8)** | 3.28 |
> | Auto-SNR | **80.9 (0.7)** | 49.9 (0.2) | **93.5 (0.2)** | 77.3 (0.3) | 17.3 (0.7) | 86.3 (0.7) | 78.5 (1.8) | 4.14 |
> | Auto-RGN | 81.4 (0.6) | **51.2 (0.2)** | **93.5 (0.3)** | **80.6 (1.2)** | **87.7 (2.8)** | **88.0 (0.7)** | 82.2 (2.7) | **1.29** |
>
> [1] Jeremy Howard and Sebastian Ruder. Universal language model fine-tuning for text classification. ACL, 2018.
>
> [2] E. Miguel Romero, Yannet Interian, Timothy Solberg, and Gilmer Valdes. Targeted transfer learning to improve performance in small medical physics datasets. Medical Physics, 2020.
>
> [3] Ananya Kumar, Aditi Raghunathan, Robbie Matthew Jones, Tengyu Ma, and Percy Liang. Finetuning can distort pretrained features and underperform out-of-distribution. ICLR, 2022.
>
>
> > Not clear how the method of surgically fine-tuning a network would compare to existing methods for domain adaptation or methods that perform test-time adaptation (adjusting the covariate shift, etc.)
>
> Our original submission included a comparison to test-time adaptation: Section 2.2 combines a state-of-the-art online test-time adaptation model (MEMO, NeurIPS 22, [4]) with surgical fine-tuning. Tables 1, 2, and Figure 4 show that surgical fine-tuning improves test-time adaptation in the online setting, where the model retains updates from past test images. We also emphasize that the main contribution of our paper is not in proposing a new method but in uncovering and analyzing a counterintuitive phenomenon.
>
> [4] Zhang, Marvin, Sergey Levine, and Chelsea Finn. "Memo: Test time robustness via adaptation and augmentation." arXiv preprint arXiv:2110.09506 (2021).
>
> > Is the reported metric for CIFAR-C and IN-C (1- mean corruption error (mCE)), in Tables 1,2, and 3, and Figures 2, 3 and 4? Because mCE is a standard metric to track when dealing with corruption datasets.
>
> We have added mCE metrics on the ImageNet-C dataset (Table 5). We reproduce the table below:
> | Method    | mCE |
> | ----------- | ----------- |
> | Vanilla ResNet-50 (no adaptation) | 76.7 |
> | Cross-Val (oracle) | 61.7 |
> | Full Fine-tuning | 62.8 |
> | Gradual Unfreeze (First->Last) | 62.4 |
> | Gradual Unfreeze (Last->First) | 63.6 |
> | L1 Regularize | 64.7 |
> | Auto-SNR | 62.9 |
> | Auto-RGN | **61.9** |
> These results give similar conclusions to the average corrupted accuracies reported in Table 3. Nevertheless, we would like to clarify that our paper's primary focus is presenting a counterintuitive phenomenon rather than achieving the state-of-the-art in transfer learning benchmarks.

---

> ### Author Response · Authors · 2022-11-16
> **Following up post-rebuttal**
>
> We wanted to follow up to see if the response and revisions address your concerns. We would be happy to provide further clarifications and revisions if you have any further questions. Thank you again for all of your hard work reviewing!

---

> > ### Comment · Reviewer_eWpB · 2022-11-18
> > **Thank you for the responses**
> >
> > Thank you authors for the responses. The authors have addressed my concerns. With additional results included in the paper, I'm comfortable increasing my score from 5 to 6.

---

### Author Response · Authors · 2022-11-13
**General Response to All Reviewers and Summary of Revisions**

We thank all the reviewers for their thoughtful and thorough comments. All reviewers appreciated the findings in the paper, and we believe that the feedback received has also helped improve the paper.

Based on the feedback, we ran additional experimental comparisons. Our changes to the text can be found in the revised pdf in maroon. In summary, our key changes include:
- Comparisons to two additional fine-tuning methods in Table 3 (as suggested by eWpB).
- Additional experiments on real-world datasets with severe distribution shifts (Camelyon and FMoW) using larger models (as suggested by QAba and Y4ag). We find that even in these settings with natural shifts, surgical fine-tuning with the first layers matches or outperforms full fine-tuning.

We hope that these additional experimental results have addressed the concerns of the reviewers. If there are any remaining questions, please let us know!

---

### Decision · Program_Chairs · 2023-01-20

**Decision:**

Accept: poster

**Justification For Why Not Higher Score:**

The work is good, but I would prefer to reserve spotlights for works that represent (i) either a more unique perspective on an understudied aspect of the ICLR community, or (ii) a more substantial breakthrough either theoretical or in capabilities of an ML model.  This paper represents a step forward on a clear path the community already follows.

**Justification For Why Not Lower Score:**

The paper gives valuable advice supported by theoretical and empirical evidence of strategies for fine tuning, which is a common step in our community.  It would eventually be useful to select one of the strategies in a given experimental setup and cite this paper as a justification of the experimental design.

**Metareview: Summary, Strengths And Weaknesses:**

The submission analyzes different forms of fine-tuning, restricting to certain parts of a network, and its relationship to shifts in data distribution.  The reviewers were unanimous in their opinion of the suitability of the study for publication at ICLR.  The contributions include a systematic labeling of different forms of data shift, theoretical analysis of fine-tuning strategies, and empirically supported recommendations for fine tuning methods.  Reviewers were initially concerned that the experiments were too small scale, and did not include large amounts of dataset shift.  These concerns were alleviated by additional results during the rebuttal/discussion phase, resulting in an overall increase in appreciation of the reviewers.  Additional concerns about code availability were addressed and the authors have promised to release it upon acceptance.  Overall, the submission addresses a relevant topic in the practice of NN fine tuning, and provides sufficient theoretical and empirical analysis to support their findings.

**Note From Pc:**

if the above contains the word "oral" or "spotlight" please see: "oral" presentation means -> notable-top-5% and "spotlight" means -> notable-top-25%. As stated in our emails, we are disassociating presentation type from AC recommendations